# Primitive Vision: Improving Diagram Understanding in MLLMs

**Shan Zhang** [1 2]   **Aotian Chen** [3]   **Yanpeng Sun** [4]   **Jindong Gu** [5]   **Yi-Yu Zheng** [6]   **Piotr Koniusz** [2 7]   **Kai Zou** [6]
**Anton van den Hengel** [1 8]   **Yuan Xue** [3]

## Abstract

Mathematical diagrams have a distinctive structure. Standard feature transforms designed for natural images (*e.g.*, CLIP) fail to process them effectively, limiting their utility in multimodal large language models (MLLMs). Current efforts to improve MLLMs have primarily focused on scaling mathematical visual instruction datasets and strengthening LLM backbones, yet fine-grained visual recognition errors remain unaddressed. Our systematic evaluation on the visual grounding capabilities of state-of-the-art MLLMs highlights that fine-grained visual understanding remains a crucial bottleneck in visual mathematical reasoning (GPT-4o exhibits a 70% grounding error rate, and correcting these errors improves reasoning accuracy by 12%). We thus propose a novel approach featuring a geometrically-grounded vision encoder and a feature router that dynamically selects between hierarchical visual feature maps. Our model accurately recognizes visual primitives and generates precise visual prompts aligned with the language model's reasoning needs. In experiments, PRIMITIVE-Qwen2.5-7B outperforms other 7B models by 12% on MathVerse and is on par with GPT-4V on MathVista. Our findings highlight the need for better fine-grained visual integration in MLLMs. Code is available at github.com/AI4Math-ShanZhang/SVE-Math.

## 1. Introduction

Diagrams are critical to many forms of communication, from corporate reports to news media. Their construction is both an art and a science (see (Tufte, 1991) for a particularly insightful exposition of the art). Despite their importance in human communication, current visual feature transforms frequently misinterpret diagrams, often hindering overall model performance in the greater task.

The visual information represented in diagrams is crucial in mathematical problem-solving. When humans perform reasoning, visual thought facilitates detailed reasoning and textual thought supports logical reasoning (Lehmann et al., 2010). While Large Language Models (LLMs) have demonstrated impressive capabilities in textual mathematical reasoning (Yu et al., 2023; Ying et al., 2024; Azerbayev et al., 2023), their proficiency often diminishes when tasks require integrating visual data (a.k.a., Multimodal Large Language Models (MLLMs)). Recent advances in MLLM (Zhang et al., 2024b; Shi et al., 2024; Kazemi et al., 2023) rely mainly on constructing large-scale mathematical visual instruction datasets, which are costly and labor-intensive to create and often involve the use of advanced models like GPT-4o (OpenAI, 2023b) to generate diverse prompts for synthetic datasets. Despite extensive effort, they still struggle to perceive and ground basic geometric primitives in mathematical diagrams.

We systematically analyzed MLLMs' ability to describe geometric entities using a meticulously collected set of 100 images from the Geo170K dataset (Gao et al., 2023a). We then manually reviewed its responses to categorize the correct descriptions and error types. As demonstrated in Fig. 1a, we observed that GPT-4o misperceived visual information in approximately 70% of cases involving geometric entities. Correcting these visual perception errors led to a 12% overall accuracy improvement on corresponding mathematical questions (refer to Fig. 5a in the Appendix). This finding highlights that misunderstanding of geometric primitives, such as lines, circles, angles, boundaries, and junctions, remains a critical bottleneck in the mathematical reasoning capabilities of MLLMs.

To mitigate above challenges, we propose a novel PRIMITIVE (**PRI**mitives in the **M**athemat**I**cal in**T**erpretat**I**on of **V**isually **E**ncoded information) , which shifts the focus from scaling mathematical visual instruction datasets to enhancing fine-grained visual perception. Specifically, we train

---

[1]Australian Institute for Machine Learning [2]Data61♥CSIRO [3]The Ohio State University [4]National University of Singapore [5]University of Oxford [6]NetMind.ai [7]Australian National University [8]The Commonwealth Bank of Australia. Correspondence to: Anton van den Hengel , Yuan Xue <yuan.xue@osumc.edu>.

*Proceedings of the 42$^{nd}$ International Conference on Machine Learning*, Vancouver, Canada. PMLR 267, 2025. Copyright 2025 by the author(s).

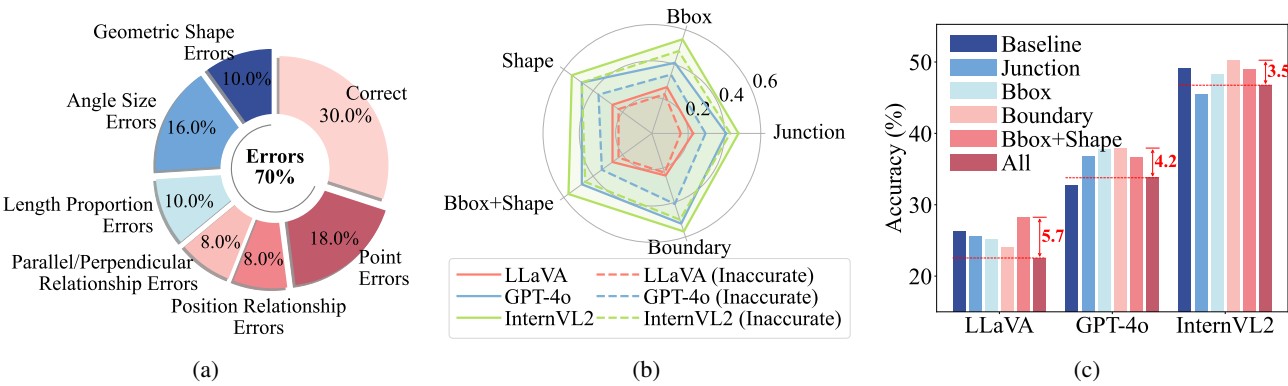

Figure 1: Performance of MLLMs on mathematical visual reasoning tasks from GeoQA test set. GPT-4o misperceives geometric entities in 70% of cases (Fig. 1a). Providing optimal geometric information improves performance, while excessive visual cues reduce top-1 accuracy—even below the baseline for text-only questions (Fig. 1c). Model performance is highly sensitive to visual cue accuracy, with a -13.6% drop in GPT-4o's top-1 accuracy when given incorrect bounding box locations and shape names (Bbox+Shape) (Fig. 1b).

an auxiliary visual encoder, namely GeoGLIP (Geometric-Grounded Language-Image Pre-training), tailored to recognize geometric primitives. While existing mathematical datasets lack bounding box and pixel-level annotations, generating such training data is simple yet highly efficient, *e.g.*, through the Matplotlib Python library. Furthermore, training protocols for visual-centric tasks are more straightforward than those for MLLMs.

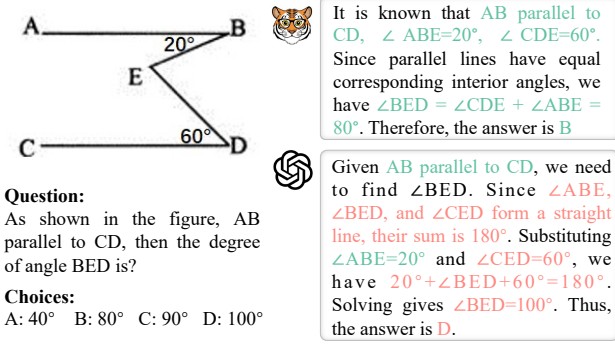

**Question:**
As shown in the figure, AB parallel to CD, then the degree of angle BED is?
**Choices:**
A: 40°   B: 80°   C: 90°   D: 100°

▷ GPT-4o struggles to correctly interpret mathematical elements, leading to errors in reasoning and relational descriptions in LLMs. By integrating GeoGLIP, PRIMITIVE effectively grounds geometric elements and their spatial relations (*e.g.*, ∠CDE), enabling more accurate reasoning. See the Appendix for additional examples.

Incorporating PRIMITIVE enables MLLMs to identify the critical visual components of mathematical problems before engaging in reasoning. Our hypothesis and specific designs are inspired by observations illustrated in Fig. 1b and Fig. 1c. Instructing MLLMs with fine-grained visual information, such as junction points and object locations, improves top-1 accuracy compared to providing only worded questions. However, excessive visual cues reduce accuracy (-4.2% in

GPT-4o), emphasizing the importance of relevance (see § A.5 for a case study). Performance is also sensitive to visual cue accuracy, with significant drops when given randomly generated incorrect coordinates. Given GeoGLIP's inherent uncertainty in detecting geometric primitives, instead of directly prompting LLMs with primitive locations (*e.g.*, hard coordinates; see §A.4 for ablation), we leverage global pyramid feature maps that encode essential information for pixel-to-shape detection. These visual cues are dynamically selected by the designed feature router, generating the so-called visual soft prompts.

We evaluate PRIMITIVE on several public mathematical benchmarks, and experimental results demonstrate its superior performance compared to models of the same or even larger sizes. Specifically, our model outperforms other 7B-parameter models and achieves comparable results to advanced 13B-parameter MLLMs, all while using a smaller dataset for visual training (40K) and 60K + 110K for alignment and instruct learning, compared to the 588K + 834K dataset used in MAVIS (Zhang et al., 2024b). These results highlight the effectiveness of our approach and underscore the importance of precise visual perception in mathematical visual reasoning. Our contributions are as follows:

- We design GeoGLIP to directly tackle the root cause of geometrical visual recognition errors in mathematical reasoning tasks by enhancing the visual encoder to perceive geometric primitives. Compared with large-scale mathematical instruction datasets, the training data generation process is simple yet highly efficient.

- We design a connector mechanism that effectively integrates the relevant geometric visual information into the language model, boosting performance without altering the reasoning components.

- GeoGLIP integrates seamlessly with diverse LLM backbones without modifying their reasoning components. Extensive experiments demonstrate that PRIMITIVE outperforms existing models of comparable and larger sizes on math benchmarks.

## 2. Related Work

**Multimodal Large Language Models for Mathematics.** Large Language Models (LLMs) have garnered significant attention, with much research focused on text-based mathematical problem-solving, expanding mathematical datasets and utilizing data augmentation (Yu et al., 2023; Yue et al., 2023b; 2024; Luo et al., 2023). Meanwhile, advancements in vision-language alignment models, such as CLIP (Radford et al., 2021) and BLIP (Li et al., 2022a), have significantly progressed multimodal tasks, leading to the development of Multimodal Large Language Models (MLLMs) (Bai et al., 2023; Gemini Team, 2023; Ye et al., 2023a; Lin et al., 2023; Gao et al., 2024; Hu et al., 2024). With the rise of instruction-following LLMs, LLaVA (Liu et al., 2024b) projects visual tokens into LLMs using a linear layer, while MiniGPT-4 (Zhu et al., 2023) reduces computation by re-sampling visual tokens into fixed-length tokens.

Building on these advancements, researchers have explore visual mathematical problem-solving using MLLMs. Unified frameworks like UniGeo (Chen et al., 2022a), Uni-Math (Liang et al., 2023), and GeomVerse (Kazemi et al., 2023) expand multimodal mathematical datasets and improve MLLM performance in geometry and diverse tasks. Leveraging current datasets, G-LLaVA (Gao et al., 2023a) constructed the Geo170K dataset, enhancing geometric problem-solving and surpassing GPT-4V (OpenAI, 2023b) on the MathVista benchmark (Lu et al., 2023) with only 7B parameters. GeoGPT4V (Cai et al., 2024a) further improved model performance on MathVista and MathVision (Wang et al., 2024) by creating a high-quality geometric problem dataset using GPT-4 and GPT-4V. MAVIS (Zhang et al., 2024b) specializes in mathematical tasks with a three-stage training pipeline including a math-specific vision encoder, while Math-LLaVA (Shi et al., 2024) introduced MathV360K, a large-scale dataset with high-quality images and diverse question-answer pairs to improve multimodal mathematical reasoning. These math-specific MLLMs have shown promising performance across several benchmark datasets (Yue et al., 2023a; Zhang et al., 2024a).

Despite these advancements, MLLMs still face challenges in multimodal mathematical tasks, particularly due to limitations in visual perception (Sun et al., 2025). While CLIP remains a common choice for many mathematical MLLMs and is known to benefit multimodal tasks, its limitations have also been identified. For instance, (Tong et al., 2024) examines 'CLIP-blind pairs', revealing that visually distinct images are often misinterpreted as similar, highlighting systematic shortcomings in CLIP's visual perception. These findings underscore the need for more specialized visual encoding methods tailored to mathematical contexts, as well as more rigorous evaluations of MLLMs' visual capabilities.

**Open-Set Object Detection.** Open-set object detection identifies arbitrary classes using existing bounding box annotations and language generalization. Methods like OV-DETR (Zareian et al., 2021), ViLD (Gu et al., 2022), Det-CLIP (Yao et al., 2022), and Grounding DINO (Liu et al., 2024d) integrate language models with detection frameworks to improve category-specific detection. However, these models often struggle with small-scale object detection due to insufficient fine-grained visual understanding. GLIP (Li et al., 2022b) addresses this limitation by integrating textual information with visual region features early in the pipeline via a language-aware deep fusion mechanism, enhancing region-level embeddings. GLIP improves detection of smaller objects and demonstrates strong zero-shot capabilities. While GLIP's potential has been explored in various fields (Surís et al., 2023; Peng et al., 2023; Li et al., 2023), its application to mathematical reasoning, particularly in precise geometric entity description and fine-grained detail identification in mathematical diagrams, remains largely unexplored. Our work extends these concepts, developing a geometric-grounded language-image pre-training model (GeoGLIP) tailored for the unique demands of visual mathematical reasoning.

**Junction and Boundary Detection.** Junction and boundary detection are crucial in object recognition (Dollar et al., 2006; Maire et al., 2008; Parida et al., 1998), and can play a pivotal role in mathematical reasoning with geometric diagrams. Junctions represent points where lines intersect, and boundaries delineate object shapes. Traditional methods like Canny edge detection (Canny, 1986) and the Hough Transform (Duda & Hart, 1972) struggle with complex diagrams and fine-grained details required for accurate mathematical reasoning. Recent deep learning approaches, such as junction detection networks (Huang et al., 2018), detect key points by considering surrounding regions. Boundary detection models like Field of Junctions (FoJ) (Verbin & Zickler, 2021) use a bottom-up approach with 'generalized M-junctions' to detect contours and junctions.

## 3. Methods

### 3.1. Overview

PRIMITIVE integrates visual understanding of geometric primitives with textual analysis to enhance the model's capability in solving mathematical problems involving visual elements. As illustrated in Fig. 2, our pipeline builds upon the LLaVA-1.5 (Liu et al., 2023b) architecture (refer to §A.1),

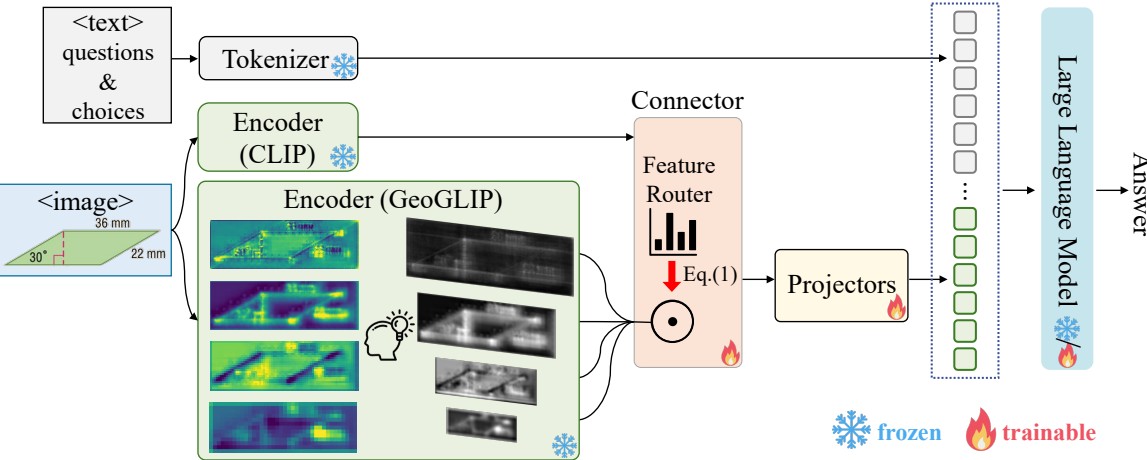

Figure 2: The diagram presents the architecture of PRIMITIVE, highlighting key innovations in the geometric-grounded vision encoder (GeoGLIP) and the feature router. Fine-grained visual understanding is achieved through a feature pyramid comprising $F_{geo}^{1*}$, $F_{geo}^{3}$, $F_{geo}^{4}$, $F_{geo}^{5}$ (attention maps displayed on the left), capturing hierarchical visual features essential for pixel-to-shape detection, respectively. The feature router dynamically adjusts the contribution of these features to generate visual soft prompts. These prompts are then combined with CLIP visual tokens and textual inputs before being fed into the language model (LLM), enabling accurate visual perception and enhanced mathematical reasoning.

introducing key innovations in the GeoGLIP and visual feature connector (additional implementations with DeepSeek-Math-7B-Instruct (Shao et al., 2024), and Qwen2.5-Math-7B-Instruct (Yang et al., 2024) are also provided). Feature maps from different layers of the GeoGLIP encoder are processed through the connector, where a feature router optimally integrates the feature pyramid into visual soft prompts by leveraging geometric information. These visual prompts are then fused with CLIP vision tokens, either along the sequence dimension or the channel dimension, and aligned with text embeddings via projection layers for visual understanding. Since channel-wise fusion offers better computational efficiency and comparable performance to sequence-based fusion in our experiments, we set channel-wise fusion as the default approach.

### 3.2. Geometric-Grounded Language-Image Pre-training

Our proposed GeoGLIP extends GLIP (Li et al., 2022b) to perform shape grounding, boundary and junction detection tasks with no human annotations. The architecture of GeoGLIP is shown in Fig. 8 of the Appendix. For shape grounding, we follow the same pipeline structure as the original GLIP model for bounding box detection (refer to §A.2 for pipeline details) but train it on the mathematical domain. The training datasets are discussed in Sec. 3.4.

The feature pyramids in the visual encoder capture different levels of diagram information: higher-resolution features are responsible for fine-grained pixel detection, while lower-resolution features capture semantic information necessary for shape detection. To further enhance fine-grained visual

perception, we use a cross-resolution mixture to integrate low-resolution features into high-resolution ones, aiding boundary and junction detection (as discussed below).

**Boundary and junction detection.** GLIP-T utilizes Swin-Tiny as its backbone, producing a five-level feature pyramid $\{F_{geo}^{i}\}_{i \in \{1,2,3,4,5\}}$, where each level's resolution is progressively downscaled by a factor of 2. To enrich the high-resolution features with semantic information, we first pass the high-resolution tensor $F_{geo}^{2}$ (as the Query) and the low-resolution tensor $F_{geo}^{4}$ (as the Key and Value) to a Multi-Head Self Attention (MHSA) module. The resulting feature maps are upsampled by a factor of 2 and element-wise added to $F_{geo}^{1}$, producing $F_{geo}^{1*}$. The rationale behind this design is to fully integrate the hierarchical object concepts at various scales produced by the downsampling layers with the high-resolution spatial information encoded by the initial embedding layer. Taking $F_{geo}^{1*}$ as input, we then adopt two decoders for boundary and junction detection (see Fig. 9).

The boundary decoder consists of two successive perception blocks, each comprising an upsampling operation using nearest-neighbor interpolation, followed by a $3 \times 3$ convolution (Conv2d), batch normalization (BN2d), and ReLU activation. The final output is resized to the original image resolution using bilinear upsampling. A junction represents the intersection of lines, determined by the intersection coordinates and the orientations of the lines. Accordingly, our junction decoder has two branches. The first branch estimates the confidence of a junction falling within each grid cell of the original image (using a $60 \times 60$ grid) and its relative position to the cell's center coordinates. The second

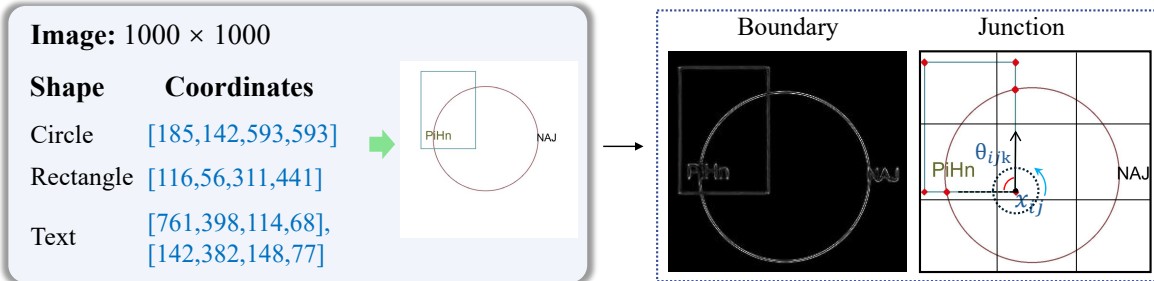

Figure 3: Process for generating synthetic data with box- and pixel-level annotations, used to tranin our GeoGLIP visual encoder. 'Text' is a random string of alphanumeric characters with a length between 1 and 10, placed alongside other geometric objects, *i.e.*, circles and rectangles. Refer to Fig. 6 in the Appendix for the detailed flow chart.

branch predicts the orientations of the intersecting lines and their confidence in falling into one of 15 evenly spaced bins within each grid cell, where each bin covers 24 degrees, ensuring the full 360-degree range is divided evenly (15 bins × 24 degrees = 360 degrees). In the junction decoder, the input $F_{\text{geo}}^{1^*}$ is first processed through a perception block, where it is upsampled to a resolution of $60 \times 60$. Then, two separate Conv2D units predict the cell confidence and location, with output sizes of $60 \times 60 \times 1$ and $60 \times 60 \times 2$, respectively. Additionally, two other Conv2D units predict the bin confidence and orientation, both producing outputs of $60 \times 60 \times 15$. For further details, refer to training step 1 in §A.6.1 and the pipeline in Fig. 9 of the Appendix.

### 3.3. Connector Design

Selecting key visual cues enhances mathematical visual problem-solving, while redundant information can hinder it. To manage feature contributions, we propose a dynamic feature router $R$, implemented as a Multi-Layer Perceptron (MLP). The router takes spatially averaged pooled feature maps from each level of GeoGLIP ($\bar{F}_{\text{geo}}^i \in \mathbb{R}^{1 \times 256}$) and the CLIP feature map ($\bar{F}_{\text{clip}} \in \mathbb{R}^{1 \times 1,024}$) as input, calculating routing weights for each feature ($\{\boldsymbol{w}^i\}_{i \in \{1,2,3,4\}} \in \mathbb{R}^{1 \times 4}$). This serves as a soft router (Puigcerver et al., 2024), with alternative router types like sparse and constant routers discussed in Sec. 4. The soft router's process is defined as:

$$\widehat{F}_{\text{geo}}^i = \mathbf{w}^i \cdot MLP \odot \mathcal{G} \odot F_{\text{geo}}^i, \quad \mathbf{w}^i = \sigma \odot R([\bar{F}_{\text{geo}}^i, \bar{F}_{\text{clip}}]), \tag{1}$$

where $F_{\text{geo}}^i$ is resized ($\mathcal{G}$) to match the spatial dimensions of $F_{\text{clip}}$ and processed by an MLP to align its channel dimensions. The symbol $\odot$ denotes the operation flow from right to left. The scalar routing weights $\mathbf{w}^i$ are then applied to the respective features. Depending on the fusion strategy with $F_{\text{clip}}$, the final $\widehat{F}_{\text{geo}}$ is generated either by element-wise addition of the weighted features $\widehat{F}_{\text{geo}} = \sum_{i=1}^{4} \widehat{F}_{\text{geo}}^i$, where the weights $\mathbf{w}^i$ are normalized ($\sigma$) using the function Soft-Max (*i.e.*, $\sum_{i=1}^{4} \mathbf{w}^i = 1$), or by channel-wise concatenation of the weighted features, where the weights are processed through a Sigmoid function.

Next, we explore strategies for fusing the soft prompts $\widehat{F}_{\text{geo}}$ with $F_{\text{clip}}$, either sequence-wise or channel-wise. In the sequence-wise method, additional tokens are added after the CLIP tokens, extending the sequence length. In contrast, channel-wise fusion combines all visual tokens along the channel dimension, maintaining the same sequence length. To enable the subsequent

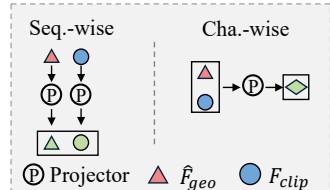

LLM to understand these visual components, the fused visual tokens are then fed into projection layers, which project the visual modality into the LLM's embedding space. Following the LLaVa-1.5 approach, we employ highly effective MLP projectors (linear layer + GELU + linear layer, a.k.a., `mlp2x_gelu`) for this task. In the sequence-wise approach, two separate projectors are applied for CLIP and soft prompts, respectively. For example, the projection matrices for the two linear layers, per projector, $\boldsymbol{\Phi_1}$ and $\boldsymbol{\Phi_2}$, have sizes of $1,024 \times 4,096$ and $4,096 \times 4,096$, where $4,096$ corresponds to the text embedding dimension. In the channel-wise approach, a single projector ($\boldsymbol{\Phi_1} \in \mathbb{R}^{5,120 \times 4,096}$ and $\boldsymbol{\Phi_2} \in \mathbb{R}^{4,096 \times 4,096}$) is used to process the combined visual tokens.

### 3.4. Training Samples for Visual-centric GeoGLIP

To enable GeoGLIP to perceive fine-grained mathematical elements, we supervise its training using datasets with box- and pixel-level annotations. The model is trained with a classical detection loss $\mathcal{L}_{det}$ (Eq. 2), a junction loss $\mathcal{L}_{junc}$ (Eq. 3), and a boundary loss $\mathcal{L}_{bodr}$ (the $\ell_2$ loss between predicted heatmap values and ground truth values). The detection loss $\mathcal{L}_{det}$ is applied to the shape grounding task, using synthetic images and FigureQA (Kahou et al., 2018) training data annotated with bounding boxes and shape names (left panel of Fig. 3) . These annotations are stored in a COCO-style JSON file for seamless integration with standard GLIP. See §A.3 for details on the synthetic data engine and dataset statistics (Figures 5b and 5c).

For boundary and junction detection tasks, we leveraged off-the-shelf models (Huang et al., 2018; Verbin & Zickler, 2021) to extract junctions and boundaries as ground truth. In addition to our synthetic samples, we incorporated the public dataset Geo170K (Chen et al., 2021b) and generated the corresponding ground truth. Specifically, junction labels include intersection coordinates and line orientations. As noted, each grid cell and bin are responsible for predicting the coordinates and the orientations, and we have $60 \times 60$ cells&15 bins per cell. The labels are formatted as $JP_{ij} = (x_{ij}, c_{ij}, \{\theta_{ijk}, c^{\theta}_{ijk}\}^{K}_{k=1})$, where $x_{ij}$ denotes the junction center coordinates, $c_{ij} \in \{0, 1\}$ indicates the presence of a junction, $\theta_{ijk}$ is the angle of the $k$-th bin, and $c^{\theta}_{ijk} \in \{0, 1\}$ is the indicator for that bin (right panel of Fig. 3).

## 4. Experiments

### 4.1. Experimental Setup

**Implementation details.** Our work follows a structured three-stage training pipeline, including multi-task visual perception training for GeoGLIP, visual-language alignment, and mathematical instruction tuning for MLLMs (refer to §A.6.1 for training & implementation details).

In summary, for multi-modal training, we freeze GeoGLIP encoder in Stage 2 and train only the projection layers to align diagram-language pairs. In Stage 3, we unfreeze both projection layer and LLM for instruction-following tuning. Our GeoGLIP together with a pre-trained CLIP ViT-L are integrated into the language models LLAMA-2, DeepSeek-Math-7B-Instruct, and Qwen2.5-Math-7B-Instruct. Images are resized to $448 \times 448$ pixels for CLIP and $1000 \times 1000$ for GeoGLIP. We train PRIMITIVE for one epoch for cross-modal alignment and two epochs for instruction tuning on Geo170K (Gao et al., 2023a), evaluating on GeoQA. For fair comparison, we train on MathV360k (Shi et al., 2024) with a batch size of 16 for one epoch, evaluating on Math-Vista (Lu et al., 2023) and MathVerse (Zhang et al., 2024a).

**Evaluation benchmarks and metric.** We evaluate PRIMITIVE on three public benchmarks: MathVerse (Zhang et al., 2024a), GeoQA (Gao et al., 2023a), and MathVista (Lu et al., 2023). MathVerse focuses on multi-modal reasoning, combining text and diagrams. GeoQA emphasizes geometric reasoning, and MathVista includes diverse tasks, *e.g.*, IQTest, PaperQA, and IconQA, covering various problem-solving domains. The predicted answers are compared to ground truths to determine top-1 accuracy, following the respective dataset protocols.

### 4.2. Main Results

Table 1 presents the comparison results on the testmini set of MathVerse, where PRIMITIVE-7B achieves state-of-the-art performance among all models using LLaMA2-7B as the base LLM, with a 5.5% improvement and achieves comparable top-1 accuracy to the most powerful open-source LLaVA-NeXT (8B) (Liu et al., 2024a) despite being smaller in size (19.3% *vs*. 21.2%). When using DeepSeek-Math-7B-Instruct (Shao et al., 2024) as the base LLM, our model's performance further increases by an additional +3.1%. Notably, even on the challenging MathVista benchmark, our model outperforms the advanced SPHINX-Plus-13B (Gao et al., 2024), and is on par with close-sourced GPT-4V (OpenAI, 2023b), as shown in Table 2. These results highlight the critical role of fine-grained visual perception in advancing mathematical reasoning capabilities in MLLMs. Tables 3 and 4 report our model's performance on plane geometry and function analysis tasks, respectively. Compared to the second-best model, MAVIS (Zhang et al., 2024b), which is trained on an $8\times$ larger mathematical visual instruction dataset, PRIMITIVE with LLaMA2-7B demonstrates superior reasoning and generalization capabilities. As menonstrated by these comparisons, constructing large instruction datasets for training MLLMs is labor-intensive and costly, whereas synthetic datasets for training visual-centric tasks offer a more scalable and efficient alternative.

**Baseline Comparison.** The effectiveness of geometric soft visual prompts is validated through a comparison between PRIMITIVE-7B and G-LLaVA across Tables 1-3. Both models utilize the same LLM (LLaMA2-7B) and instruction training dataset, ensuring a controlled evaluation. PRIMITIVE-7B demonstrates clear improvements with +7.7% on MathVerse +12.3% on MathVista, and +2.8 % on GeoQA. To further evaluate generalizability and effectiveness across different LLMs, we extend the comparison to two additional base models, *e.g.*, Deepseek-Math-7B and Qwen-Math-7B, on the MathVista benchmark, as shown in Table 5b, where '(–)' denotes variants without geometric soft visual prompts.

The above variants are math-specific models trained solely on mathematical text-diagram pairs (MathV360K & Geo170K). In contrast, generic models like Qwen2.5-VL-7B are pre-trained on large-scale multimodal instruction data (2TB+), enabling broader reasoning capabilities. To assess the effectiveness of our method in a generalist setting, we perform an ablation study on Qwen2.5-VL-7B. Specifically, we fine-tune both the projector and LLM using LoRA on MathV360K & Geo170K. Under this setup, integrating soft visual prompts yields performance gains, whereas direct fine-tuning slightly reduces performance (Table 5c).

### 4.3. Ablation Analysis

**Hierarchical feature maps & feature router types.** The lowest-resolution feature maps $F^{5}_{\text{geo}}$ (semantic-rich) assist in shape grounding (Fig. 4a), while highest-resolution feature maps $F^{1}_{\text{geo}}$ (geometric-rich) aid pixel-level boundary

Table 1: **Results on testmini set of MathVerse** with the accuracy metric. The highest results for closed-source and open-source MLLMs are highlighted in red and blue respectively.

| Model | Base LLM | All | Text Dominant | Text Lite | Vision Intensive | Vision Dominant | Vision Only |
|---|---|---|---|---|---|---|---|
| | | Acc | Acc | Acc | Acc | Acc | Acc |
| *Baselines* | | | | | | | |
| Random Chance | - | 12.4 | 12.4 | 12.4 | 12.4 | 12.4 | 12.4 |
| Human | - | 67.7 | 71.2 | 70.9 | 61.4 | 68.3 | 66.7 |
| *LLMs* | | | | | | | |
| ChatGPT (Ouyang et al., 2022) | - | 26.1 | 33.3 | 18.9 | - | - | - |
| GPT-4 (OpenAI, 2023a) | - | 33.6 | 46.5 | 46.5 | - | - | - |
| *Closed-source MLLMs* | | | | | | | |
| Qwen-VL-Plus (Bai et al., 2023) | - | 11.8 | 15.7 | 11.1 | 9.0 | 13.0 | 10.0 |
| Gemini-Pro (Gemini Team, 2023) | - | 23.5 | 26.3 | 23.5 | 23.0 | 22.3 | 22.2 |
| Qwen-VL-Max (Bai et al., 2023) | - | 25.3 | 30.7 | 26.1 | 24.1 | 24.1 | 21.4 |
| GPT-4V (OpenAI, 2023b) | - | 39.4 | 54.7 | 41.4 | 34.9 | 34.4 | 31.6 |
| *Open-source MLLMs* | | | | | | | |
| LLaVA-1.5 (Liu et al., 2023a) | Vicuna-13B | 7.6 | 8.8 | 7.6 | 7.4 | 7.4 | 6.9 |
| SPHINX-Plus (Gao et al., 2024) | LLaMA2-13B | 12.2 | 13.9 | 11.6 | 11.6 | 13.5 | 10.4 |
| SPHINX-MoE (Gao et al., 2024) | Mixtral-8×7B (Jiang et al., 2024) | 15.0 | 22.2 | 16.4 | 14.8 | 12.6 | 9.1 |
| G-LLaVA (Gao et al., 2023a) | LLaMA2-7B | 16.6 | 20.9 | 20.7 | 17.2 | 14.6 | 9.4 |
| InternLM-XC2. (Dong et al., 2024) | InternLM2-7B (Cai et al., 2024b) | 16.5 | 22.3 | 17.0 | 15.7 | 16.4 | 11.0 |
| ShareGPT4V (Chen et al., 2023b) | Vicuna-13B | 13.1 | 16.2 | 16.2 | 15.5 | 13.8 | 3.7 |
| Math-LLaVA (Shi et al., 2024) | Vicuna-13B | 19.0 | 21.2 | 19.8 | 20.2 | 17.6 | 16.4 |
| LLaVA-NeXT (Li et al., 2024) | LLaMA3-8B (Meta, 2024) | 19.3 | 24.9 | 20.9 | 20.8 | 16.1 | 13.8 |
| **PRIMITIVE-7B** | LLaMA2-7B | 21.2 | 26.4 | 23.2 | 22.9 | 18.0 | 15.4 |
| **PRIMITIVE-Deepseek-7B** | Deepseek-math-7B (Shao et al., 2024) | 24.3 | 31.1 | 26.9 | 25.6 | 19.3 | 17.5 |
| **PRIMITIVE-Qwen2.5-7B** | Qwen-math-7B (Academy, 2023) | 28.5 | 36.9 | 34.1 | 31.4 | 25.1 | 15.2 |

Table 2: **Results on testmini set of MathVista** with the accuracy metric. The highest results for closed-source and open-source MLLMs are highlighted.

| Model | Base LLM | All | FQA | GPS | MWP | TQA | VQA |
|---|---|---|---|---|---|---|---|
| | | Acc | Acc | Acc | Acc | Acc | Acc |
| *Baselines* | | | | | | | |
| Random Chance | - | 17.9 | 18.2 | 21.6 | 3.8 | 19.6 | 26.3 |
| Human | - | 60.3 | 59.7 | 48.4 | 73.0 | 63.2 | 55.9 |
| *Closed-source MLLMs* | | | | | | | |
| Qwen-VL-Plus (Bai et al., 2023) | - | 43.3 | 54.6 | 33.5 | 31.2 | 48.1 | 51.4 |
| GPT-4V (OpenAI, 2023b) | - | 49.9 | 43.1 | 50.5 | 57.5 | 65.2 | 38.0 |
| *Open-source MLLMs* | | | | | | | |
| mPLUG-Owl2 (Ye et al., 2023b) | LLaMA-7B | 22.2 | 22.7 | 23.6 | 10.2 | 27.2 | 27.9 |
| MiniGPT-v2 (Chen et al., 2023a) | LLaMA2-7B (Touvron et al., 2023) | 23.1 | 18.6 | 26.0 | 13.4 | 30.4 | 30.2 |
| G-LLaVA (Gao et al., 2023a) | LLaMA2-7B | 25.1 | 19.1 | 48.7 | 3.6 | 25.0 | 28.7 |
| LLaVA-1.5 (Liu et al., 2023a) | Vicuna-13B | 27.7 | 23.8 | 22.7 | 18.9 | 43.0 | 30.2 |
| SPHINX-Plus (Gao et al., 2024) | LLaMA2-13B | 36.7 | 54.6 | 16.4 | 23.1 | 41.8 | 43.0 |
| **PRIMITIVE-7B** | LLaMA2-7B | 37.4 | 31.9 | 53.9 | 29.0 | 41.4 | 30.8 |
| **PRIMITIVE-Deepseek-7B** | Deepseek-math-7B (Shao et al., 2024) | 48.7 | 37.6 | 63.0 | 48.7 | 48.1 | 35.8 |
| **PRIMITIVE-Qwen2.5-7B** | Qwen-math-7B (Academy, 2023) | 50.4 | 38.7 | 67.3 | 58.1 | 51.2 | 31.8 |

detection but still fail to capture fine details (Fig. 4b). Our cross-resolution mixture method, producing $F_{\text{geo}}^{1^*}$, achieves superior fine-detail perception (Fig. 4c). See §A.4 for a detailed analysis and more variants. We then examine three types of routers: constant, sparse, and the default soft router $R$ (Tab. 6a). The constant router assigns equal weights $w^i = 0.25$ to each $F_{\text{geo}}^i$, while the sparse router selects one feature map of GeoGLIP with $w^i \in \{0, 1\}$. As expected, in

Table 3: Comparison of geometric numerical answer accuracies (%) on **GeoQA**.

| Model | Accuracy (%) |
|---|---|
| Random Chance | 25.0 |
| Frequent Guesses | 32.1 |
| *Top-10 Accuracy* | |
| NGS (Chen et al., 2021a) | 56.9 |
| DPE-GPS (Cao & Xiao, 2022) | 62.7 |
| SCA-GPS (Ning et al., 2023) | 64.1 |
| *Top-1 Accuracy* | |
| Geoformer (Chen et al., 2022b) | 46.8 |
| UniMath (Liang et al., 2023) | 50.0 |
| G-LLaVA (Gao et al., 2023a) | 64.2 |
| MAVIS-7B (Zhang et al., 2024b) | 66.7 |
| **PRIMITIVE-7B** | 67.0 |
| **PRIMITIVE-Deepseek-7B** | 72.8 |
| **PRIMITIVE-Qwen2.5-7B** | 79.6 |

Table 4: Comparison of model performance on **FunctionQA of MathVista**.

| Model | Accuracy (%) |
|---|---|
| Random Chance | 22.5 |
| *Closed-source MLLMs* | |
| CoT GPT-4 (OpenAI, 2023a) | 35.0 |
| PoT GPT-4 (OpenAI, 2023a) | 37.0 |
| Multimodal Bard (Google, 2023) | 45.5 |
| GPT-4V (OpenAI, 2023b) | 69.5 |
| *Open-source MLLMs* | |
| LLaVA (Liu et al., 2023b) | 20.5 |
| LLaMA-Adapter V2 (Gao et al., 2023b) | 32.0 |
| LLaVA-NeXT (Liu et al., 2024a) | 33.7 |
| SPHINX-MoE (Gao et al., 2024) | 34.6 |
| MAVIS-7B (Zhang et al., 2024b) | 40.3 |
| **PRIMITIVE-7B** | 40.5 |
| **PRIMITIVE-Deepseek-7B** | 45.1 |
| **PRIMITIVE-Qwen2.5-7B** | 53.3 |

Table 5: Comparison of top-1 accuracies (%) on GeoQA w.r.t. different visual encoder variants (Table 5a). Table 5b shows top-1 accuracy on testmini set of MathVista w.r.t. with (PRIMITIVE) or without our soft visual prompts (PRIMITIVE (-)) across different LLM base models. Table 5c shows the instantiation of our method on the Qwen2.5-VL-7B.

| Type | Encoders | Accuracy (%) |
|---|---|---|
| Dual encoders | GLIP+CLIP | 65.3 |
| Dual encoders | GeoGLIP+CLIP | **67.0** |
| Single encoder | GeoGLIP | 66.1 |
| Single encoder | CLIP | 64.2 |

(a)

| Model | Base LLM | Acc (All) |
|---|---|---|
| G-LLaVA | LLaMA2-7B | 25.1 |
| PRIMITIVE | LLaMA2-7B | 37.4 |
| PRIMITIVE-Deepseek(-) | DeepSeek-math-7B | 42.3 |
| PRIMITIVE-Deepseek | DeepSeek-math-7B | 48.7 |
| PRIMITIVE-Qwen2.5(-) | Qwen2.5-math-7B | 44.0 |
| PRIMITIVE-Qwen2.5 | Qwen2.5-math-7B | **50.4** |

(b)

| Model | MathVista Acc(all) | MathVerse Acc(all) |
|---|---|---|
| Qwen2.5-VL-7B | 68.2 | 49.2 |
| PRIMITIVE-Qwen2.5-VL(-) | 65.0 | 45.3 |
| PRIMITIVE-Qwen2.5-VL | **69.7** | **51.0** |

(c)

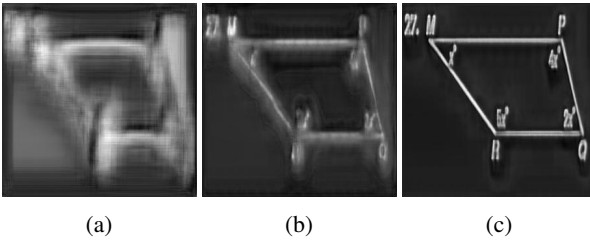

(a)          (b)          (c)

Figure 4: Hierarchical features capture different levels of geometric information, dynamically weighted by the feature router. Semantic-rich feature $F_{geo}^5$ effectively perceives geometric shapes but lacks sufficient resolution for boundary localization (Fig. 4a), while geometric-rich feature $F_{geo}^1$ enhances pixel-level boundary detection (Fig. 4b). The cross-resolution feature $F_{geo}^{1*}$ captures fine-grained visual cues (Fig. 4c), enhancing boundary detection.

the sparse router, $F_{geo}^{1*}$ with rich visual information, achieves the highest accuracy (hierarchical attention maps in Fig. 2). The soft router outperforms the others, demonstrating its effectiveness for dynamic routing of multiple signals.

**Necessity of CLIP & Impact of visual encoder variants.** We present a comprehensive analysis of visual encoder vari-

Table 6: Ablation results w.r.t. top-1 accuracy on GeoQA. Tab. 6a shows results for feature router types; Results for connector designs are shown in Fig. 6b.

| | Seq.-wise | Cha.-wise |
|---|---|---|
| Constant $R$ | 63.9 | 62.8 |
| Sparse $R$ | $F_{geo}^{1*} \rightarrow$ 64.2 $\quad \rightarrow$ 64.9 | |
| | $F_{geo}^3 \rightarrow$ 61.1 $\quad \rightarrow$ 61.8 | |
| | $F_{geo}^4 \rightarrow$ 61.9 $\quad \rightarrow$ 62.3 | |
| | $F_{geo}^5 \rightarrow$ 61.9 $\quad \rightarrow$ 61.6 | |
| Soft $R$ | **66.6** | **67.0** |

(a)

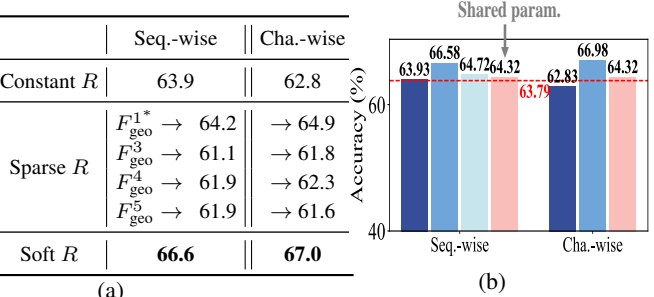

(b)

ants, with a summary of their impact shown in Table 5a. We designed a variant that excludes the CLIP visual encoder, relying solely on our soft prompts from the GeoGLIP visual encoder. This resulted in an accuracy drop from 67.0% to 66.1%, though it still outperformed the CLIP encoder alone (64.2%). These results demonstrate that while CLIP lacks fine-grained perception, its general visual features still benefit text-visual alignment in MLLM training.

**Impact of math-specific fine-tuning for GeoGLIP.** We

leveraged the hierarchical pyramid features from the GLIP visual encoder (fine-trained on natural image datasets, such as MS COCO). The results are shown in Table 5a. To ensure a fair comparison, we utilized feature maps with the same resolution: the first layer with the largest resolution and the last three layers with smaller resolutions. This resulted in a performance drop from 67.0% to 65.3%, with only a minimal +1.1% gain over G-LLaVA. The slight boost likely stems from integrating high-resolution vision features, which are not sensitive to geometric details, as GLIP fails to detect basic geometric shapes (Fig. 10). We also fine-tune DETR-based Grounding DINO (Liu et al., 2024c) as GeoDINO, reducing top-1 accuracy on GeoQA from 67.0% to 66.1% (vs. GeoGLIP). See §A.4 for analysis.

**The robust of GeoGLIP.** To assess the robustness of PRIMITIVE in junction and boundary detection, we evaluated it on a custom set of 500 mathematical diagrams with various visual distortions. Standard metrics from natural image tasks are adopted: junction detection is measured by recall with a confidence threshold of 0.65, and boundary detection by Intersection over Union (IoU), computed from binarized maps (threshold=200) via pixel-wise logical AND/OR between predictions and ground truth. We applied the following distortions: 1) **Gaussian Noise:** Added with a variance of 0.3 to simulate noisy conditions; 2) **Resolution change:** Reduced the shortest image dimension from 800 to 400 pixels; 3) **Line style modification:** Replaced solid lines with dashed lines.

As shown in Table 7, GeoGLIP exhibits strong robustness to resolution and line width variations, attributed to training-time augmentations such as randomized line width, horizontal flipping, center cropping, and ratio-preserving resizing. The model is more sensitive to Gaussian noise, with a 4.6% drop in junction recall and 3.1% in boundary IoU. Dashed lines increase false positive junction detections, resulting in a 3.2% drop in performance. These results highlight both the strengths and limitations of the current system and motivate further improvements in data generation and training strategies to better handle diverse visual distortions.

Table 7: Robustness of GeoGLIP under visual distortions.

| Distortion Type | Recall % | IoU % |
|---|---|---|
| None (w/o) | 85.6 | 92.3 |
| + Gaussian Noise | 81.0 | 89.2 |
| + Resolution Change | 85.2 | 91.9 |
| + Line Width Variation | 85.9 | 92.3 |
| + Dashed Lines | 82.4 | 92.7 |

**Impact of Detection Errors on Reasoning Accuracy.** To assess how detection errors affect downstream reasoning, we introduced Gaussian noise (variance=0.3) to evaluation images on the GeoQA benchmark. Table 8 reports top-1 accuracy with or without distortion. Our three variants

Table 8: Effect of Gaussian noise on GeoQA top-1 accuracy.

| Model | Acc. (w/o) | Acc. (w/ Gau. Noise) |
|---|---|---|
| PRIMITIVE | 67.0 | 65.7 |
| PRIMITIVE-Deepseek | 72.8 | 71.0 |
| PRIMITIVE-Qwen2.5 | 79.6 | 76.9 |
| PRIMITIVE (-) | 64.2 | 61.1 |
| PRIMITIVE (-)-Deepseek | 66.1 | 61.9 |
| PRIMITIVE (-)-Qwen2.5 | 72.3 | 67.2 |

show modest drops of -1.3, -1.8, and -2.7, respectively. In contrast, their CLIP-based counterparts (PRIMITIVE (-)) exhibited larger declines, indicating that CLIP encoders are more sensitive to noise and propagate less reliable visual signals to the reasoning module.

**Connector designs.** We examine the impact of the number of projection experts. The default channel concatenation setup utilizes a single expert with a `mlp2x_gelu`. In the multi-expert ablation, where two sequential `mlp2x_gelu` are applied, the top-1 accuracy drops from 66.98% to 64.32% (-2.66%), as shown in Fig.6b. For sequence-wise fusion, which uses two separate projectors by default, we ablate shared parameters across these projectors, making them act as a single-projection expert. Fig. 6b shows that the multi-expert setup enhances sequence-wise performance compared to shared parameters (a.k.a., a single expert), boosting accuracy from 64.32% to 66.58% (+2.26%). We hypothesize that the improvement in sequence-wise fusion may stem from the added flexibility in handling heterogeneous inputs, whereas in channel-wise fusion, it could introduce unnecessary complexity and redundancy.

**More ablation studies.** See §A.5 for case studies and §A.6.2 for efficiency analysis.

# 5. Conclusion

We mitigate the limitations of current mathematical MLLMs by directly tackling their deficiency in perceiving geometric primitives, which are essential for visual mathematical reasoning. We proposed PRIMITIVE, a novel approach that enhances mathematical visual reasoning by integrating a geometry-aware visual encoder trained through multi-task objectives. Our method avoids the labor-intensive process of building large-scale mathematical visual instruction datasets, offering a more efficient and scalable solution. By designing a feature router that dynamically adjusts the contribution of each visual cue, we generate soft prompts that guide the language model toward better mathematical reasoning without overwhelming it with redundant or irrelevant visual data. Extensive experiments across three public mathematical benchmarks validate the effectiveness of PRIMITIVE. We believe our work offers a new paradigm for solving mathematical problems in a visual context, emphasizing the critical role of fine-grained primitive visual grounding.

## Impact Statement

This paper presents a novel approach aimed at advancing the field of Multimodal Large Language Models (MLLMs) for mathematical reasoning tasks. Our work focuses on enhancing fine-grained visual perception through the proposed GeoGLIP framework, addressing a critical bottleneck in current MLLMs' performance. Our systematic analysis and 'apples-to-apples' comparison provide critical insights for future research, highlighting the need for more effective integration of fine-grained visual understanding in MLLMs. There are many potential societal consequences of our work, none which we feel must be specifically highlighted here.

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

# Primitive Vision: Improving Diagram Understanding in MLLMs

## A. Appendix

In this supplementary material, we illustrate the related background for our method (§ A.1), provide a detailed description for GeoGLIP (Geometric-Grounded Language-Image Pre-training) pipeline (§ A.2), explain the process of synthetic data generation, and outline the datasets used for training GeoGLIP (§ A.3), present more ablation studies of PRIMITIVE (§ A.4), offer case studies that illustrate the practical application of our feature router mechanism and chain-of-thought (CoT) reasoning results (§ A.5), demonstrate the training details/efficiency of PRIMITIVE (§ A.6) and examine our model's limitations while outlining potential directions for future work (§ A.7).

### A.1. Background

**Grounded Language-Image Pre-training (GLIP).** GLIP (Li et al., 2022b) unifies detection and grounding by reformulating object detection as phrase grounding. It accepts paired image-text inputs, where the text consists of candidate detection categories, such as the 80 COCO object class names joined by '.', *i.e.*, person. bicycle. car. $\cdots$ toothbrush. In GLIP, object classification logits in the box classifier (traditional object detection) are replaced with word-region alignment scores, computed as the dot product between region visual features and phrase language features. GLIP operates as a two-stage detector, composed of: 1) A Swin Transformer as a visual encoder, which extracts features $F_I$ of images $X_I$ and passes $F_I$ to a Region Proposal Network (RPN) to generate region coordinates, and then corresponding region features $O_I$ are cropped from $F_I$; 2) A pre-trained BERT model as the language encoder, to embed the input text $X_L$ into token embeddings $P_L$; 3) A language-aware deep fusion module $\text{Fus}_{IL}$ that fuses $O_I$ and $P_L$ in the last few encoding layers. The final alignment scores $S_{\text{ground}}$, calculated as:

$$O_I = \text{RPN}(\text{Swin}(X_I)), \quad P_L = \text{BERT}(X_L), \quad O'_I, P'_L = \text{Fus}_{IL}(O_I, P_L) \quad S_{\text{ground}} = O'_I, P'^{\top}_L.$$

**Large Language and Vision Assistant (LLaVA).** We adopt (Large Language and Vision Assistant) LLaVA's architecture (Liu et al., 2023b) as the basis. LLaVA leverages the complementary strengths of pre-trained large language models and visual encoders to perform multi-modal tasks, consisting of a large language model $f_\phi$ (Vicuna (Chiang et al., 2023)), a vision encoder (CLIP, ViT-L/14) (Radford et al., 2021), and a projection layer. The projection layer projects the visual embedding from the vision encoder into the text embedding space. LLaVA begins by processing an input image $X_I$ through the CLIP visual encoder, which extracts visual features $F_I = \text{CLIP}(X_I)$. To bridge the gap between the image features and the language model's word embedding space, LLaVA applies a simple linear projection matrix $\mathbf{\Phi}$, converting visual features $F_I$ into visual tokens $H_I$, which are compatible with the language embedding space:

$$H_I = \mathbf{\Phi} \cdot F_I, \text{ with } F_I = \text{CLIP}(X_I)$$

The visual tokens $H_I$ and language instruction tokens $P_L$ are passed into the language model for joint reasoning and language generation as $f_\phi([H_I, P_L])$.

### A.2. GeoGLIP

The GeoGLIP pipeline is shown in Fig. 8, where the RPN and language-aware deep fusion details are omitted for clarity. The GeoGLIP takes image-text paired as input: an image containing geometric shapes and a text listing the shape classes (*i.e.*, 'circle. trapezoid. triangle. ... line.'). These inputs are processed by the GeoGLIP encoder, which generates feature pyramids at multiple scales ($F_{\text{geo}}^1, F_{\text{geo}}^2, F_{\text{geo}}^3, F_{\text{geo}}^4, F_{\text{geo}}^5$). Each feature pyramid contains different levels of detail, capturing varying levels of geometric information. These features are then routed to three separate detectors: 1) Shape Detector: identifies and localizes basic geometric shapes by generating bounding boxes for objects within the image; 2) Junction Detector: detects junctions or intersections of geometric entities in the image; 3) Boundary Detector: identifies boundaries of geometric shapes, refining their outlines for more accurate representation. The combination of the feature pyramids with task-specific detectors allows GeoGLIP to perform fine-grained visual tasks in a mathematical context.

In Fig. 9, we illustrate detailed designs about junction and boundary detectors:

- Junction Detector: The detector processes the feature $F_{\text{geo}}^{1^*}$ through a decoder, identifying the confidence of junction points within each grid cell and their relative positions. It also predicts the orientations and confidence levels of intersecting lines within the grid, split into multiple angular bins to cover the 360-degree range.

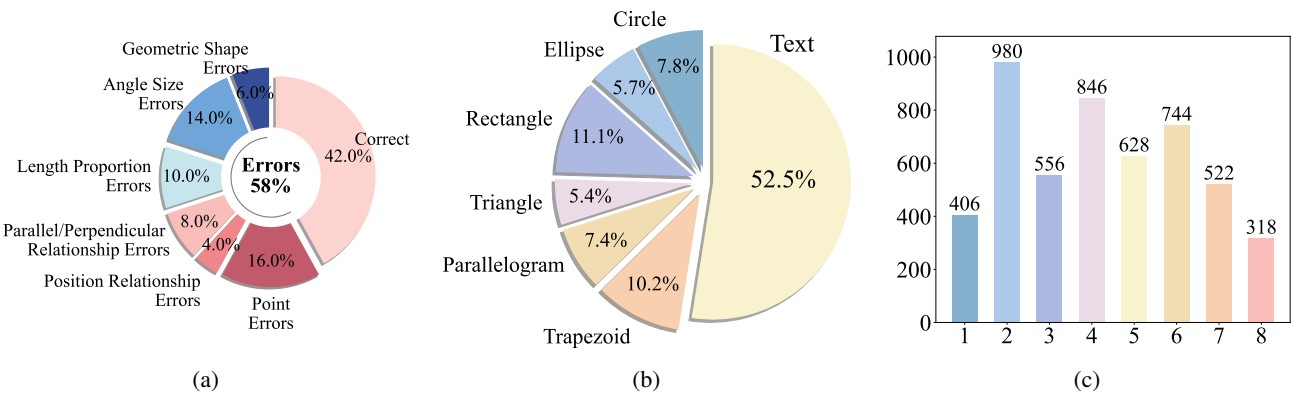

(a)                                    (b)                                    (c)

Figure 5: Fig. 5a presents the statistics of top-1 accuracy after manually correcting the visual perception errors shown in Fig.1a of the main paper, which initially caused incorrect answers to mathematical questions. Specifically, we restated the output of GPT-4o w.r.t. each type of visual recognition error and calculated the accuracy of its answers. Overall, correcting these visual perception errors led to an approximate 12% increase in accuracy on the corresponding mathematical questions. Fig. 5b and Fig. 5c present the data statistics for synthetic math-specific datasets, including the distribution of geometric shapes/classes and the number of objects per image. Each geometric object has a 70% probability of being assigned an alphanumeric text, leading to a higher proportion of the 'Text' class.

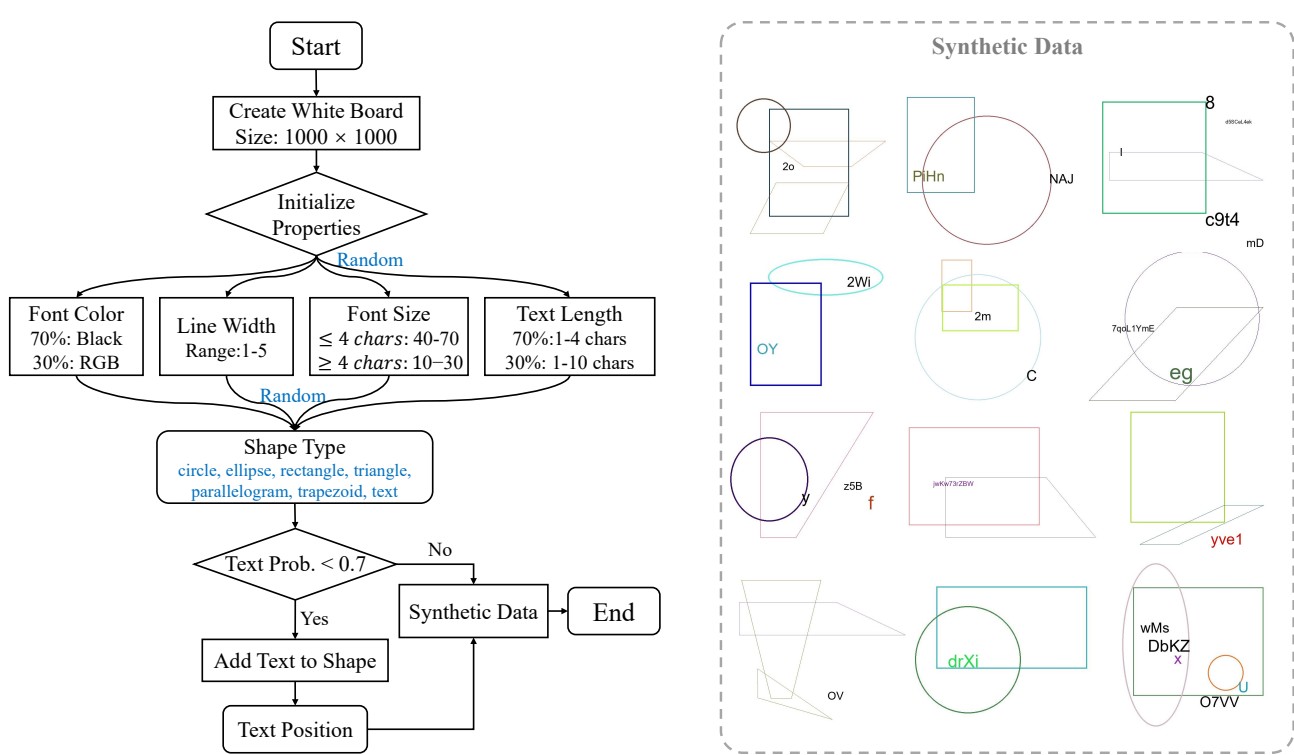

Figure 6: The flow diagram depicts the process for generating synthetic math-specific datasets, along with visualizations of the generated data samples.

- Boundary Detector: It employs two successive perception blocks and upsampling operations to restore the feature map to the original image resolution for boundary decoding.

Both detectors use multi-resolution feature maps from the GeoGLIP encoder, and specific design for each task is optimized to capture relevant geometric properties, contributing to enhanced mathematical visual reasoning. Refer to § 3.2 of main paper for more details.

## A.3. Training Dataset for GeoGLIP

Notably, our synthetic math-specific datasets differs from the traditional mathematical instruction datasets, and we do not create or use any additional self-generated instruction datasets beyond the publicly available Geo170K (Gao et al., 2023a) and MathV360K (Shi et al., 2024) datasets for MLLM training. Instead, our synthetic samples, annotated with box/pixel-level details, are exclusively utilized to train the GeoGLIP. Compared to constructing mathematical instruction datasets, our synthetic data generation process is significantly more efficient and resource-friendly. It does not require manual labeling, as all data can be programmatically generated, *e.g.*, through the Matplotlib Python library. In contrast, constructing instruction datasets often relies on GPT-4o to create diverse prompts and necessitates human intervention, making the process labor-intensive and costly.

**Shape grounding.** To generate *synthetic datasets* for object grounding tasks, we employ an automated Python-based approach that efficiently creates images containing geometric shapes and text with associated bounding boxes, class labels, and annotations. The geometric categories include shapes like circles, ellipses, rectangles, triangles, parallelograms, trapezoids, and text. A variable number of basic geometric shapes and alphanumeric text elements are generated, with font sizes dynamically adjusted according to text length. These shapes are randomly distributed within a $1000 \times 1000$ pixel canvas, while text is positioned either inside or adjacent to the shapes with a 70% probability. Bounding boxes are then calculated for each shape and text element, ensuring they remain within image bounds. Finally, shapes and text are assigned class labels and coordinates, saved in a COCO-style JSON file for seamless integration with standard GLIP. Fig. 6 shows the detailed flow diagram. Fig. 5b and Fig. Fig. 5c present the data statistics for synthetic math-specific datasets, including the distribution of geometric shapes and the number of objects per image. In addition to 10,000 synthetic images, we incorporated 20,672 images from the *FigureQA* training dataset with bounding box annotations for the shape grounding task.

**Junction and boundary detection.** We utilized off-the-shelf models (Huang et al., 2018; Verbin & Zickler, 2021) to extract junctions and boundary as ground truth on both our *synthetic dataset* and public *Geo170K* training images (9,426 diagrams). We then designed junction and boundary heads, parallel to the object detection head, with all tasks sharing the same visual encoder. Through this multi-task learning approach, our GeoGLIP can perceive rich visual information in the mathematical domain.

## A.4. Quantitative analysis

**GeoGLIP detection visualizations.** Fig. 10 illustrates shape detection results on Geo170K, FigureQA and our synthetic test dataset, while Fig. 11 presents the results for boundary and junction detection. Our detector successfully localizes basic geometric shapes and junction points while providing pixel-level boundary results in most cases. However, in complex scenarios such as overcrowded or occluded settings, the detector may struggle. Moreover, in junction detection, some failure cases involve numerous detections but with low accuracy. This issue arises due to noisy ground truth during the training phase, as manually labeling junctions is tedious and time-consuming. To address this, we use an off-the-shelf model (Huang et al., 2018) to generate ground-truth labels for junction detection. However, since this model was trained on images of man-made environments, it faces an out-of-domain challenge when applied to geometric objects, resulting in labels that are not fully accurate. Improving the accuracy of these labels would significantly enhance junction detection performance.

**Effect of cross-resolution mixture.** We designed four additional variants to demonstrate the effectiveness of our cross-resolution mixture approach. Recall that we have five feature levels $\{F_{\text{geo}}^i\}_{i \in \{1,2,3,4,5\}}$ with different resolutions, each with different resolutions, ranging from geometric-rich to semantic-rich information. The cross-resolution mixture aims to generate the input $F_{\text{geo}}^{1^*}$ for the boundary and junction decoders, with the expectation that $F_{\text{geo}}^{1^*}$ captures more informative visual information to benefit boundary and junction detection tasks.

Using boundary detection as an example, we first used the semantic-rich $F_{\text{geo}}^5$ as input to the boundary decoder. As shown in Fig. 7a, the decoder fails to generate clear boundaries, resulting in a blurred output. Next, we used the geometric-rich $F_{\text{geo}}^1$, which performs better (Fig. 7b), showing some visible boundaries. To further enhance the results, we applied a cross-resolution attention mechanism (classic Multi-Head Self-Attention, MHSA) between $F_{\text{geo}}^2$ and $F_{\text{geo}}^4$, improving boundary detection as seen in Fig. 7d. Since boundary detection benefits from geometric-rich information, we upsampled the cross-correlated features by a factor of 2 and added them element-wise with $F_{\text{geo}}^1$, producing the best visualization results, especially for finer details (Fig. 7e). Finally, to assess the importance of cross-resolution attention, we replaced it with element-wise addition. As expected, the boundaries became blurred (Fig. 7c) due to the reduced receptive field. Replacing addition with the attention mechanism yields similar boundary results but decreases shape grounding performance from 95.3% to 92.4% mAP on our synthetic test set. Therefore, our mixture process integrates both cross-resolution attention and

Figure 7: Qualitative boundary visualization results. Semantic-rich features with the lowest resolution lead to blurred boundaries (Fig. 7a), while geometric-rich features with the highest resolution improve clarity (Fig. 7b). The cross-resolution mixture yields the best results (Fig. 7e), compared with using either element-wise addition (Fig. 7c) or MHSA alone (Fig. 7d). Zoom in for best view.

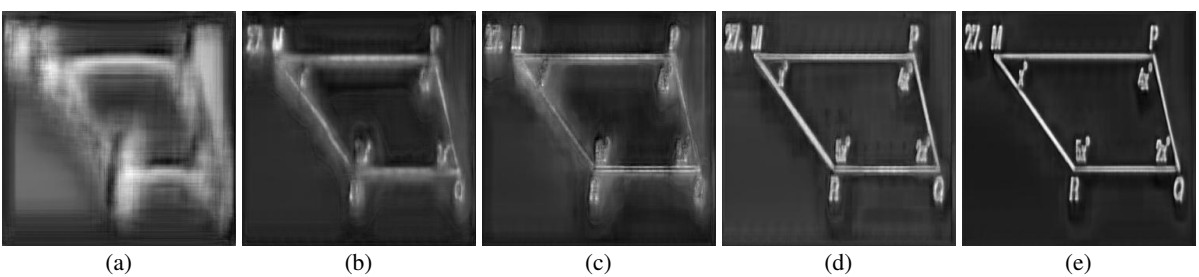

| | | | | |
|:---:|:---:|:---:|:---:|:---:|
| (a) | (b) | (c) | (d) | (e) |

addition operations.

**Necessity of GLIP architecture.** GLIP is an open-set object detector capable of identifying arbitrary classes by matching visual features with corresponding language embeddings. Unlike traditional object detectors with learnable classification weights, GLIP's multi-modal architecture offers greater generality to novel objects and surpasses previous traditional object detectors. To evaluate alternatives, we replaced GLIP with another open-set object detector, Grounding DINO (Liu et al., 2024c), and fine-tuned it on our math-specific dataset. Experimentally, we found that Grounding DINO struggles to effectively detect small-scale geometric primitives. We hypothesize that this limitation stems from architectural differences. Grounding DINO, as a DETR-based detector, relies exclusively on the last-layer features of its visual encoder for cross-attention with query embeddings during final detection. In contrast, GLIP, as a Faster-RCNN-based detector, leverages multi-scale features for both bounding box regression and classification, enabling superior small-object detection capabilities. When integrating the fine-tuned Grounding DINO encoder into our pipeline, the top-1 accuracy on the GeoQA benchmark dropped from 67.0% to 66.1%, further supporting GLIP's advantages for our tasks.

**Instructing LLMs with coordinates of primitives.** We have conducted experiments for directly providing geometric-relevant information to the model. Since no existing mathematical instruction datasets include detailed location information for geometric objects (*e.g.*, bounding box coordinates or junction points), we generated this data by inferring Geo170K training images using GeoGLIP to extract the relevant location information. This information was appended to the special token $\langle image \rangle$ as supplementary descriptions for each image, using instructions such as: "there is a bounding box at $\langle x, y, w, h \rangle$ or there is a junction at $\langle x, y \rangle$ with lines directions $\langle \theta \rangle$". When tested on the Geo170K test set of the GeoQA benchmark, the top-1 accuracy dropped from 67.0% to 63.2%. This result is close to the variant of our constant router 62.8% (assigning equal weights to all features in Table 6a). This performance drop is consistent with our systematic analysis in Fig. 1b and Fig. 1c: Inaccurate instructions would harm the performance, and relevance is key—excessive visual cues interfere with problem-solving.

### A.5. Case studies

**Selective visual information helps reasoning.** Fig. 12 showcases GPT-4o's responses based on additional visual information from geometric primitives, alongside the question, choices, and diagram $\langle image \rangle$ as inputs. We provide hard-coded coordinates for bounding boxes and junctions using instructions such as: "there is a bounding box at $\langle x, y, w, h \rangle$ (the normalized center point and width/height)" with shape names $\langle geometric\ shape \rangle$ (if shape information is provided), or "candidate junction point $\langle x, y \rangle$. For boundary information, we use "$\langle boundary\ image \rangle$ is the boundary sketch related to the main diagram" as instructions. The right side visualizes the provided visual cues in the original geometric diagram for clarity, though these images are not input into GPT-4o. Fig. 12 highlights the importance of providing relevant visual prompts for each case; otherwise, redundant information may interfere with the solving process. For example, in case 1, bounding box coordinates per object can be distracting when solving a perimeter question compared to junction locations. In contrast, pixel-level visual information (boundary) aids the model in perceiving complex geometric shapes, such as polygons and circles, and is beneficial for calculating overlap regions, while relying on junctions may lead to biased answers. In practice, selecting supporting information for each case is labor-intensive and requires the involvement of math experts. We address this challenge by using the feature router, which automatically learns which fine-grained visual information is important during the training stage.

**Response comparison.** Fig. 13 presents case studies comparing our PRIMITIVE-Deepseek-7B with GPT-4o on the MathVerse testmini set. These examples highlight the strengths of PRIMITIVE-Deepseek-7B in providing precise geometric visual information, enabling clear and logically grounded mathematical reasoning in its responses. For instance, our model demonstrates sensitivity to the positions of individual points/junctions, effectively capturing the relationships between different lines. As shown in Fig. 13a, it successfully identifies angle 1 and its relationship with angle BEF, enabling correct reasoning and answers. In contrast, GPT-4o fails to recognize these relationships, leading to flawed reasoning and incorrect answers.

Fig. 14 and Fig. 15 present a Chain-of-Thought (CoT) comparison among PRIMITIVE-Deepseek-7B, GPT-4V, and InternVL2. The results clearly demonstrate that providing geometry-aware visual cues significantly aids LLMs in understanding the relationships between geometric elements, thereby enhancing the entire reasoning process. In contrast, the other two MLLMs fail to achieve this level of understanding, leading to incorrect reasoning and outcomes. This demonstrates that without accurately recognizing visual elements, even strong LLMs struggle with reasoning tasks. As shown in GPT-4V's output, its initial misidentification of mathematical elements results in an incorrect Chain-of-Thought (CoT) response.

## A.6. Mathematical Visual Training and Efficiency

### A.6.1. TRAINING DETAILS

Our work follows a structured three-stage training pipeline, including multi-task visual perception training for GeoGLIP, visual-language alignment, and mathematical instruction tuning for MLLMs.

**Stage 1:** To enable the visual encoder in GeoGLIP to ground geometric entities in mathematical diagrams, we utilize synthetic and FigureQA training images annotated with bounding boxes for the *grounded pre-training*. Specifically, we fine-tune a pre-trained GLIP-T model (with Swin-Tiny as the backbone), adhering to the GLIP detection loss defined as:

$$\mathcal{L}_{det} = \mathcal{L}_{rpn} + \mathcal{L}_{cls} + \mathcal{L}_{reg} \tag{2}$$

where $\mathcal{L}_{rpn}$ refines the region proposals generated by the RPN, $\mathcal{L}_{cls}$ applies binary sigmoid loss to alignment scores, and $\mathcal{L}_{reg}$ uses smooth $\ell_1$ loss for bounding box regression.

Following the process in (Huang et al., 2018), for the *junction detection* task, the input image is divided into mesh grids, with each grid cell responsible for detecting a junction if its center falls within the cell. Each $ij$-th cell predicts a confidence score $c_{ij}$, indicating the likelihood of a junction in that cell. Since a junction represents the intersection of lines, the number of predictions per cell varies depending on the number of lines intersecting. To capture orientations, each cell is further divided into $K$ equal bins (default $K = 15$), with each bin spanning 24 degrees to cover the full 360-degree range. Each junction is represented as $JP_{ij} = (x_{ij}, c_{ij}, \{\theta_{ijk}, c^{\theta}_{ijk}\}^K_{k=1})$, where $x_{ij}$ denotes the junction center coordinates, $c_{ij} \in [0, 1]$ is the confidence score for the presence of a junction, $\theta_{ijk}$ is the angle of the $k$-th bin, and $c^{\theta}_{ijk}$ is the confidence score for that bin.

The loss function for junction detection consists of four terms. Given a set of ground truth junctions $JP = jp_1, \ldots, jp_N$ in an image, the loss function is formulated as:

$$\mathcal{L}_{junc} = \lambda_{loc} \cdot (\mathcal{L}^c_{loc} + \mathcal{L}^b_{loc}) + \lambda_{conf} \cdot (\mathcal{L}^b_{conf} + \mathcal{L}^b_{conf}). \tag{3}$$

The default values for the weights in Eq. 3 are $\lambda_{loc} = 0.1$ and $\lambda_{conf} = 1$, where the superscripts $c$ and $b$ refer to cell and bin, respectively. Specifically, we apply the binary cross-entropy loss for both $\mathcal{L}^c_{conf}$ and $\mathcal{L}^b_{conf}$, and use $\ell_2$ loss to measure the relative position of the predictions against the ground truth for $\mathcal{L}^c_{loc}$ and $\mathcal{L}^b_{loc}$. Refer to (Huang et al., 2018) for more details. In the *boundary detection* task, $\mathcal{L}_{bodr}$ minimizes the $\ell_2$ loss between the estimated heatmap values and the ground truth values.

Our final loss function for multi-task visual perception training is defined as:

$$\mathcal{L}_{vis} = \mathcal{L}_{det} + \mathcal{L}_{junc} + 5 \cdot \mathcal{L}_{bodr}, \tag{4}$$

where the weight for $\mathcal{L}_{bodr}$ is set to 5, while the weights for $\mathcal{L}_{det}$ and $\mathcal{L}_{junc}$ are kept at 1.

**Stage 2 & 3:** During both phases, we freeze the GeoGLIP encoder. In Stage 2, we train only the projection layers to align diagram-language pairs. In Stage 3, we unfreeze both the projection layer and the LLM to perform comprehensive

instruction-following tuning, culminating in PRIMITIVE-7B. For these two stages, we employ the conventional LLaVA loss, formulated as:

$$\mathcal{L}_{llm} = -\sum_{t=1}^{L} \log p \left[ S_{tar}^t | f_\phi(s_{tar}^{(<t)}, S_{in}, I) \right], \tag{5}$$

where $f_\phi$ denotes the model parameterized by $\phi$, $I$ corresponds to the figure, $S_{tar}$ and $S_{in}$ represent the target and input sentences, respectively; $S_{tar}^t$ refers to the $t$-th token of the target output, and $L$ denotes the sequence length.

Training is conducted on 8 A100 GPUs with a batch size of 32. The base learning rate is set to $1 \times 10^{-5}$ for the language backbone and $1 \times 10^{-4}$ for all other parameters, and it is decreased by a factor of 0.1 at 67% and 89% of the total training steps. We employ the same data augmentation strategies as GLIP, including random horizontal flipping and aspect ratio-preserving resizing with a minimum size of 800 pixels.

For multi-modal training, we freeze the GeoGLIP encoder. In Stage 2, we train only the projection layers to align diagram-language pairs. In Stage 3, we unfreeze both the projection layer and the LLM to perform comprehensive instruction-following tuning. Our GeoGLIP together with a pre-trained vision transformer (CLIP ViT-L)(Radford et al., 2021) are integrated into the language models LLAMA-2(Touvron et al., 2023), DeepSeek-Math-7B-Instruct (Shao et al., 2024), and Qwen2.5-Math-7B-Instruct (Yang et al., 2024), respectively. Images are padded to squares and resized to $448 \times 448$ pixels with a white background for processing by CLIP, and to $1000 \times 1000$ pixels for processing by GeoGLIP. We train PRIMITIVE for one epoch for cross-modal alignment and two epochs for instruction tuning on the Geo170K(Gao et al., 2023a) dataset, evaluating the model on GeoQA (Gao et al., 2023a). For fair comparison, we train our model on MathV360k (Shi et al., 2024) using a batch size of 16 for one epoch with an initial learning rate of $3 \times 10^{-5}$, evaluating on MathVista (Lu et al., 2023) and the minitest set of MathVerse (Zhang et al., 2024a).

A.6.2. EFFICIENCY

PRIMITIVE-7B introduces minimal computational overhead, as detailed in the below comparison Table 9. The GeoGLIP and Connector contribute an additional parameter size of 32.65MB and 8.73MB, and the Projectors accounting for 16.13MB. The inference time per sample increases slightly, from 19.80s to 20.04s (+0.24s). Training is conducted on 8 A100 GPUs with a batch size of 128 using the MathV360K dataset, which includes 40K images and 360K question-answer pairs. The total training time shows only a marginal increase, from 10.35h to 10.54h (+0.19h), demonstrating scalability for larger models and datasets.

Table 9: Comparison of computational overhead and parameter size for G-LLaVA and PRIMITIVE.

| #Parameter size | GeoGLIP | Connector | Projectors | Time (inference/sample) | Time (training/MathV360K) |
|---|---|---|---|---|---|
| G-LLaVA | - | - | 16.52MB | 19.80s | 10.35h |
| **PRIMITIVE** | 32.65MB | 8.73MB | 31.20MB | 20.04s | 10.54h |

A.7. Limitations and Further research

Our research aims to offer a new perspective on solving mathematical visual reasoning problems by first training a vision-centric model to provide visual prompts for LLMs, rather than focusing on creating large visual instruction fine-tuning datasets for MLLMs. Despite the effectiveness of our approach, there are several limitations to consider. First, the reliance on synthetic data for visual tasks may not fully capture the complexity of real-world geometric problems, potentially limiting generalization to more diverse datasets. Additionally, while the feature router provides automatic selection of relevant visual cues, it may not always perfectly align with human intuition or domain-specific knowledge, particularly in cases requiring more intricate reasoning.

For future research, one promising direction is to extend our method by incorporating real-world mathematical datasets to improve generalization and robustness. However, this will require some human labeling processes, as existing mathematical datasets lack detailed box or pixel-level annotations. Incorporating such annotations would provide a more accurate and fine-grained understanding of visual elements in mathematical problems, allowing models to better generalize to real-world scenarios. Developing efficient semi-automated labeling techniques or combining expert annotations with synthetic data could also help reduce the manual effort required. With improved detection performance, we may explore more advanced methods for designing soft prompts, such as object-level prompts/visual tokens. Further refinement of the feature router, such as combining it with interpretable methods to better understand its decision-making process, could also enhance

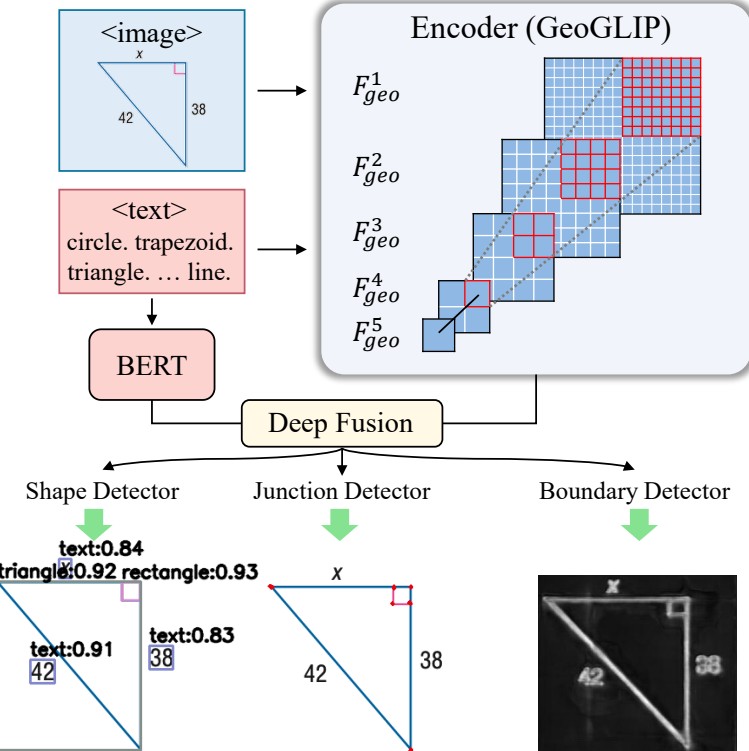

Figure 8: GeoGLIP pipeline. A geometric multi-task detector. GeoGLIP simultaneously detects multiple tasks, including basic geometric shapes, junctions, and boundaries, utilizing multi-scale features to capture fine-grained geometric entities.

the model's performance. By making the feature router more transparent, we could gain insights into how it selects and prioritizes visual cues, allowing for fine-tuning that aligns better with human intuition and task-specific requirements. This, in turn, would allow for more informed adjustments, leading to better generalization and accuracy in complex mathematical problem-solving scenarios.

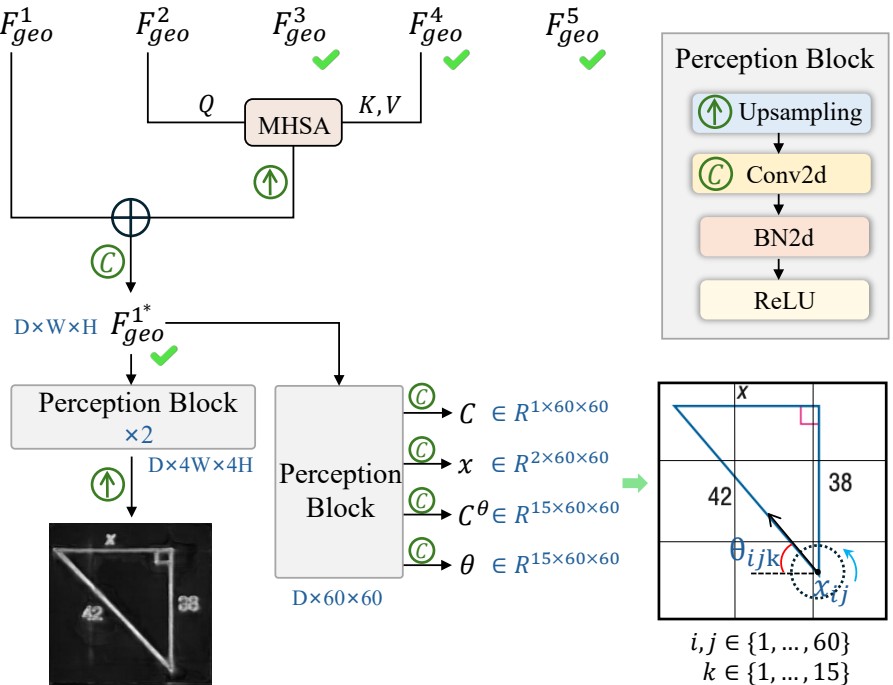

Figure 9: Designs for the junction and boundary detectors: We first use an attention mechanism (MHSA) to fuse two-scale features, followed by upsampling and addition with the highest resolution features, resulting in $F_{\text{geo}}^{1^*}$. Separate perception blocks are then applied for junction and boundary detection. For junction detection, the detector provides cell confidence ($C$), cell location ($X$), bin confidence ($C^\theta$), and bin orientation ($\theta$). Green check-marked features indicate candidate features for soft prompts, with $D, W, H$ representing channel dim., and spatial resolution (width&height).

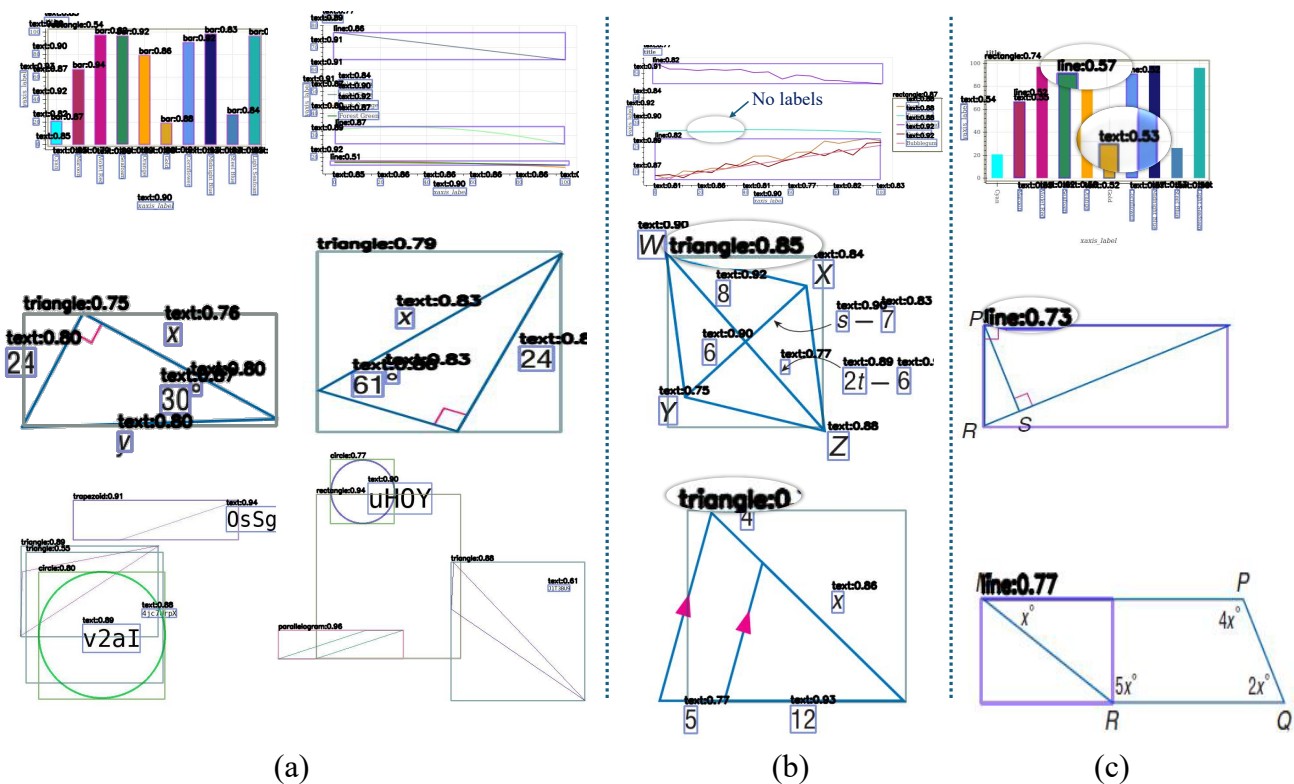

Figure 10: The visualization of shape detection on FigrueQA, Geo170K and our synthetic test dataset. The left panel (a) displays accurate shape detection results generated by GeoGLIP where even small-scale x-ticks are correctly recognized (zoom in 280% for details). GeoGLIP successfully classifies bars in histograms and rectangular shapes in geometric diagrams. The middle panel (b) represents failure cases, with all errors highlighted using a magnifying glass. For instance, in the first row figure, the cyan line is misrecognized, and three crowded lines are incorrectly grouped within a single bounding box. The results in the last panel (c) are generated by the original GLIP, trained on natural images. It is evident that most geometric shapes are misclassified as lines or text, and GLIP struggles to recognize small-scale objects, where GeoGLIP excels.

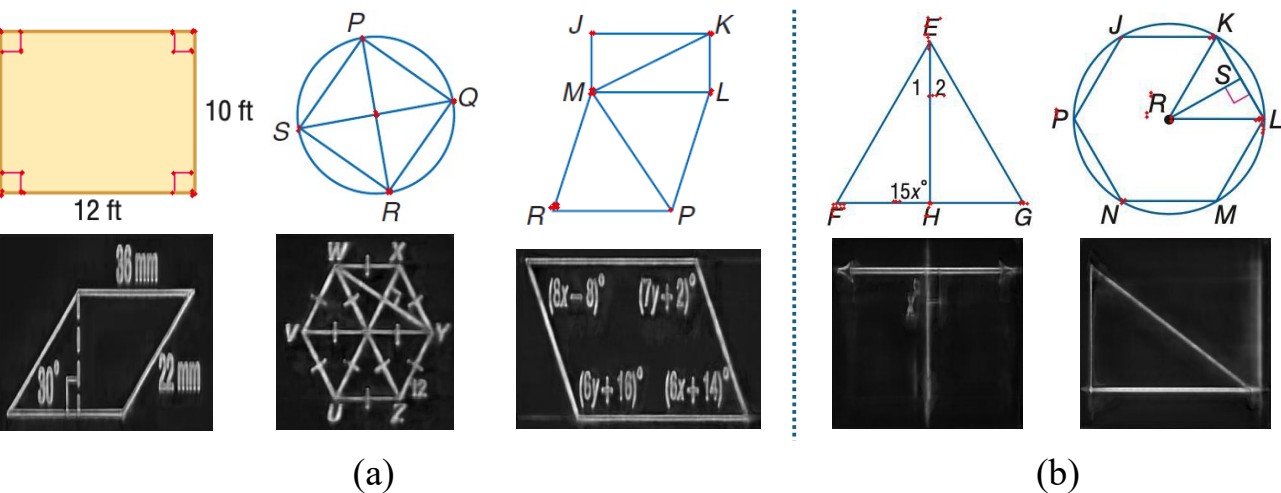

Figure 11: The visualization of junction and boundary detection results. The left panel (a) illustrates accurate detections, while the right panel (b) represents failure cases. Junction detection failures frequently exhibit redundant detections.

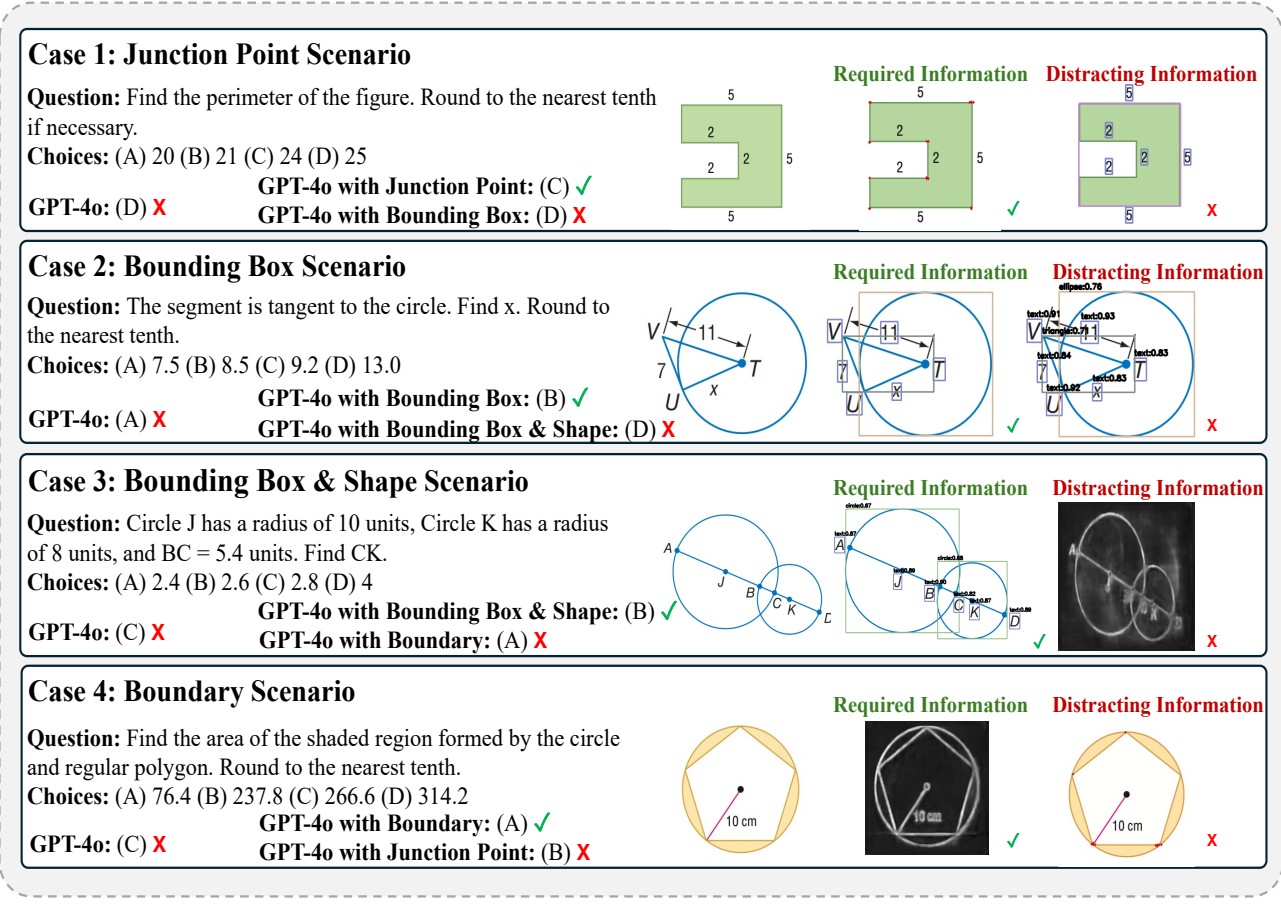

Figure 12: A case study on the Geo170K dataset (Gao et al., 2023a) highlights the importance of providing relevant visual information for each math visual question answer. Zoom in for best view.

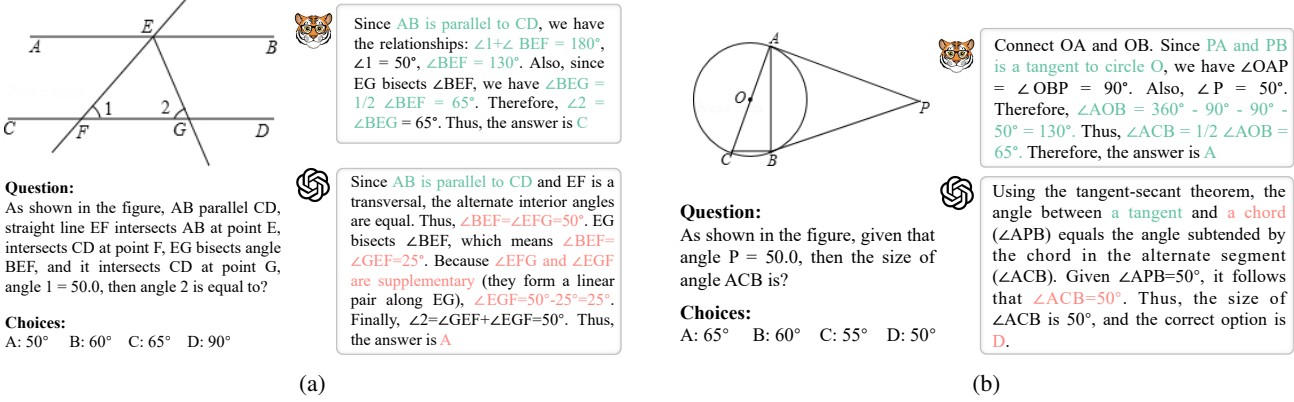

Figure 13: Response comparison of our PRIMITIVE-Deepseek-7B and GPT-4o. Refer to the main text for detailed analysis. Zoom in for best view.

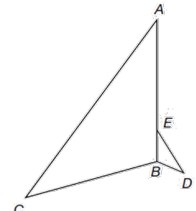

**PRIMITIVE**

✔ Since AB:EB=5:1, we have AB=5 × EB

✔ ∠A is congruent to ∠BED and ∠C is congruent to ∠D, so we can conclude that ΔABC is similar to ΔBED

✔ Therefore, the ratio of the areas of ΔABC to ΔBED is equal to the square of the ratio of AB to EB, which is (AB/EB)²=25

✔ The area of ΔABC is $25 \times (5a^2 + 10) = 125 \times a^2 + 250$ Therefore, the answer is **D**

**GPT-4V**

✔ The ratio of the length of AB to EB is given as 5:1. This means AB = 5 × EB

✘ The ΔABC and ΔEBC share the same height from point C

✘ Thus, the ratio of their areas is the same as the ratio of their bases (AB to EB), which is 5:1

✘ The area of ΔABC is $5 \times (5a^2 + 10) = 25 \times a^2 + 50$ Therefore, the answer is **B**

**InternVL2**

✔ Since AB:EB=5:1, we have AB=5 × EB

✘ E is a point on AB, and ED is parallel to AC. When a line is drawn parallel to one side of a triangle and intersects the other two sides, it forms a similar triangle. Therefore, Δ ABC is similar to ΔBED

✔ Since AB:EB=5:1, the ratio of areas will be 5²:1=25:1

✔ The area of ΔABC is $(5a^2 + 10) \times 25 = 125 \times a^2 + 250$ Therefore, the answer is **D**

**Question:**
If the ratio of the length of AB to the length of EB is 5:1, and the area of the triangle BED is $5a^2 + 10$, what is the area of triangle ABC?

**Choices:**
A: $5a^2 + 10$
B: $25a^2 + 50$
C: $25a^2 + 100$
**D: $125a^2 + 250$**
E: Cannot be determined

Figure 14: Chain-of-Thought (CoT) response comparison of our PRIMITIVE-Deepseek-7B, GPT-4V and InternVL2. Refer to the main text for detailed analysis. Zoom in for best view.

**PRIMITIVE**

✔ As shown in the figure, OB = OC, ∠ OBC = 40°, therefore ∠OCB = ∠OBC = 40°

✔ ∠ COB = 180° - 40° - 40° = 100°

✔ Therefore ∠A = 1/2 ∠COB = 50°. So the answer is A

**GPT-4V**

✘ The image shows a circle with a central angle ∠OBC, and we know that the measure of a central angle is equal to the measure of the arc it intercepts. So, the arc BC would also measure 40 degrees.

✔ So, if the arc BC is 40 degrees, then angle A, which is an inscribed angle intercepting that arc, would be half of that

✘ Therefore ∠A = 1/2 ∠OCB = 20°. However, this result does not seem to match any of the given options

**InternVL2**

✔ The figure shows a circle with center O, and points A, B, and C on the circumference. Angle OBC is given as 40 degrees.

✔ In a circle, the angle subtended by an arc at the center is twice the angle subtended at any point on the circumference on the same side of the arc. This is known as the Inscribed Angle Theorem

✘ ∠OBC is 40 degrees. ∠A is the angle subtended by the arc BC at point A. Based on the Inscribed Angle Theorem, ∠A is half of ∠OBC= 40°/ 2 = 20°. The given choices do not include 20°

**Question:**
As shown in the figure, when angle OBC = 40.0, the degree of angle A is?

**Choices:**
**A: 50°** B: 55° C: 60 D: 65°

Figure 15: Chain-of-Thought (CoT) response comparison of our PRIMITIVE-Deepseek-7B, GPT-4V and InternVL2. Refer to the main text for detailed analysis. Zoom in for best view.

