# OpenReview forum: "Primitive Vision: Improving Diagram Understanding in MLLMs"
_ICML.cc/2025/Conference — ICML 2025 poster_

### Official Review · Reviewer_sjVj · 2025-03-10

**Overall Recommendation:** 4

**Summary:**

The paper focus on the problem of math diagram understanding --- multimodal math problems with visual inputs. The paper has two main contributions, (1) GeoGLIP, a vision encoder that is more geometrically-grounded; (2) PRIMITIVE, a VLM trained with GeoGLIP + a feature selection module, which demonstrates strong performance on math diagram understanding tasks.

## Update after rebuttal
The rebuttal addresses my major concerns, I raised score accordingly; I believe adding these results are important to make the claims more convincing;

**Claims And Evidence:**

The main claims/contributions of the paper are stated as:
1. GeoGLIP feature is better than CLIP
2. The Feature Router is needed and effective

---
Although Claim 2 is well supported by ablation Table 5, Claim 1 is not very convincing.
- For Claim 1. the paper lacking important baselines that finetune the same backbone VLM, with the same data, but using their original vision encoder; this makes it unclear whether the main benefit is just mainly from the a better backbone model, or maybe better training data

**Essential References Not Discussed:**

N/A

**Experimental Designs Or Analyses:**

As mentioned in the Claims And Evidence, a set of important baseline is missing

**Methods And Evaluation Criteria:**

The method and evaluation setting are mostly clear for the claim, but some remaining questions need to be answered:

1. Additional analysis is needed for demonstrating: how robust is the GeoGLIP (especially junction and boundary detection) to the variance/noise in the visual input, for example, change of resolution, different line width, different line type (solid vs dashed), different junction size, etc. And how will the these detection error be propagated to the final performance?

2. Another concerning result is that the performance on FigureQA and VQA is not good, showing that the GeoGLIP feature might not be able to easily generalize to more diverse domains other than in-domain data as the synthesized dataset. And it is unclear that whether the performance on general multimodal benchmarks such as MME, MMBench can be retrained.

**Other Comments Or Suggestions:**

Minor: The title is exceeding the textlength, this might violate the formatting requirement, please fix that

**Other Strengths And Weaknesses:**

Some notations of the features are a little bit confusing; For example, in Figure 2's caption, it seems that the four features from GeoGLIP to Connector contains a mixed feature F1* and 3 un-mixed feature; but in Section 3.3 text, there is no notation referring to mixed feature (*); this make is unclear what exactly is the input to the Connector

**Questions For Authors:**

N/A

**Relation To Broader Scientific Literature:**

The key contributions are related to addressing the long-lasting problem in VLMs --- fine-grained visual perception and grounding.

**Theoretical Claims:**

N/A

---

> ### Author Rebuttal · Authors · 2025-04-01
>
> # We thank the reviewer for insightful questions that help refine our work further.
> ## 1. Claim 1—that GeoGLIP outperforms CLIP—is unconvincing due to missing baselines using the same VLM and data with original vision encoders, leaving it unclear whether gains come from the backbone or training data.
>
> We have conducted an ablation study in Sec. A.4  (lines 840-848 and Tab. 6), which utilizes different visual encoders trained on the same  backbone and datasets.
>
> * We compare a **single encoder** setup using either CLIP or GeoGLIP, with the same backbone VLM (LLaVA) and dataset (Geo170K). GeoQA accuracy is 64.2 for CLIP and 66.1 for GeoGLIP.
>
> * We evaluate **dual encoders** combining GLIP or GeoGLIP with CLIP, showing that without math-specific fine-tuning, GLIP's performance drops (67.0→65.3) due to limited geometric sensitivity, as illustrated in Fig. 10.
>
> If the reviewer considers this ablation crucial, we are open to moving it to the main paper.
>
> |Type|Model|Top1 Acc (GeoQA)|
> |:-:|:-:|:-:|
> |Dual encoders|GLIP+CLIP|65.3|
> |Dual encoders|GeoGLIP+CLIP|67.0|
> |single encoder|GeoGLIP|66.1|
> |single encoder|CLIP |64.2|
>
> ## 2. Additional analysis is needed to assess GeoGLIP’s robustness to visual variations (e.g., resolution, line style/width, junction size) in junction and boundary detection, and how detection errors impact final performance.
>
> In response to this concern, we evaluated the robustness of our model using a custom set of 500 images with various distortions, as no public benchmark exists for junction and boundary detection in mathematical diagrams. We adopted standard metrics from natural image tasks:
>
> * Junction Detection: Recall, using a confidence threshold of 0.65.
>
> * Boundary Detection:  Intersection over Union (IoU), with boundary maps binarized using a threshold of 200 (1 for boundary and 0 for non-boundary); IoU is computed via pixel-wise logical AND/OR between prediction and ground truth.
>
> Distortions Applied:
> * Gaussian Noise: Variance of 0.3.
>
> * Resolution Change: Shortest side reduced from 800 to 400 pixels.
>
> * Line Width Variation: Increased from 1–4 to 1–8 pixels,  correspondingly impacting junction size.
>
> * Line Style Modification: Changed from solid to dashed lines.
>
> Our results (see table) show GeoGLIP is robust to resolution and line width changes, aided by diverse training data and augmentations (e.g., varying line widths, flipping, cropping, keep-ratio resize). It is more sensitive to Gaussian noise (−4.6 junction, −3.1 boundary) and dashed lines (−3.2 junction, due to higher false positives). These results highlight the need for improved data generation and training strategies to better handle visual distortions.
>
> | Distortion Type      | Junction Detection (recall \%) | Boundary Detection (IoU \%) |
> |:-:|:-:|:-:|
> | w/o                   | 85.6                          | 92.3 |
> | +Gaussian Noise       | 81.0                           | 89.2  |
> | +Resolution Change    | 85.2                           | 91.9 |
> | +Line Width Variation | 85.9                          | 92.3|
> | +Dashed Lines         | 82.4                          | 92.7|
>
> As Gaussian noise and dashed lines are the most impactful factors, we apply Gaussian noise for further reasoning evaluation due to limitations in modifying line styles without access to the original drawing codes of evaluation images.  On GeoQA, top-1 accuracy dropped by −1.3 (PRIMITIVE-7B), −1.8 (PRIMITIVE-Qwen2.5-7B), and −2.7 (PRIMITIVE-Deepseek-7B). In contrast, the baseline model using only the CLIP encoder (PRIMITIVE(-)) showed larger drops of −3.1, −4.2, and −5.1, indicating that CLIP is more sensitive to distortions and provides less reliable visual input for reasoning.
>
> | Model                 | Top1 Acc (wo/w Gau. noise) |
> |:-:|:-:|
> | PRIMITIVE-7B          | 67.0/65.7|
> | PRIMITIVE-Deepseek-7B | 72.8/71.0 |
> | PRIMITIVE-Qwen2.5-7B  | 79.6/76.9 |
> | PRIMITIVE(-)-7B      | 64.2/ 61.1  |
> | PRIMITIVE(-)-Deepseek-7B  | 66.1/  61.9   |
> | PRIMITIVE(-)-Qwen2.5-7B | 72.3/ 67.2   |
>
> # 3. Preservation of general ability.
>
> While our model doesn't lead on FigureQA and VQA in Table 2, it significantly outperforms the G-LLaVA baseline with gains of +19.6 and +2.1, respectively. To address concerns about general ability, we further evaluated PRIMITIVE-7B on SEED-I and MM-Bench. On SEED-I, it maintained comparable performance to LLaVA-1.5-7B (66.2 vs. 66.9), and on MM-Bench, it achieved a +0.6 improvement. Most gains are observed in categories like instance interaction, counting, and spatial localization.
>
> # 4. Inconsistent notations and formatting issue.
>
> We apologize for the confusion. The four features transferred from GeoGLIP to the Connector include one mixed feature ($F^{1*}$) and three unmixed features. We will clarify this in Sec. 3.3, including revising the index notation in Eq. (1). We will adjust the title to meet the formatting requirements.

---

> > ### Comment · Reviewer_sjVj · 2025-04-03
> >
> > (copy the official comment here so that authors can see) The rebuttal addresses my major concerns, I will raise score accordingly; I believe adding these results are important to make the claims more convincing;

---

> > > ### Author Response · Authors · 2025-04-03
> > >
> > > Esteemed Reviewer,
> > > \
> > > \
> > > Thank you for your kind message, and valuable comments helping us improve and refine our manuscript. Meantime, if there is anything else we can answer or explain or discuss further, kindly do let us know.
> > > \
> > > \
> > > Rest assured, all requested explanations and additional results will be added in the final paper.
> > >
> > > Kind regards,
> > > \
> > > Authors

---

### Official Review · Reviewer_nRX6 · 2025-03-12

**Overall Recommendation:** 3

**Summary:**

This paper addresses a significant challenge in multi-modal large language models (MLLMs), specifically focusing on their limited ability to accurately interpret geometric primitives (e.g., points, lines, boundaries, and junctions) in mathematical diagrams. The authors conduct a comprehensive analysis revealing that existing models such as GPT-4o frequently misinterpret these visual elements, negatively impacting the accuracy of subsequent reasoning tasks. To address this, the authors propose PRIMITIVE, a novel approach featuring a dedicated geometric-aware visual encoder (GeoGLIP) and a dynamic feature router. GeoGLIP utilizes a multi-task learning framework based on Mask R-CNN architectures to effectively detect boundaries and junctions. The feature router dynamically selects and integrates multi-scale features from GeoGLIP and the CLIP encoder, resulting in improved reasoning performance on several benchmarks, including MathVerse, GeoQA, and MathVista. Experimental results demonstrate substantial improvements in accuracy compared to existing methods, emphasizing the importance of precise geometric visual understanding in mathematics-related tasks.

**Claims And Evidence:**

- The paper clearly identifies a crucial limitation in current MLLMs regarding precise geometric visual perception, presenting a thorough and systematic analysis of existing model failures.
- Although the authors claim to perform an 'apples-to-apples' comparison, the evaluation lacks a detailed analysis of how different visual encoders affect the reasoning performance when keeping the LLM weights fixed. For example, a comparison between Qwen2.5-vl and PRIMITIVE Qwen2.5-7B, analyzing both the performance differences and qualitative differences in responses, would provide deeper insights.

**Essential References Not Discussed:**

N/A

**Experimental Designs Or Analyses:**

- Comprehensive experimental validation across multiple challenging mathematical visual reasoning benchmarks showcases clear and convincing performance advantages over established baseline methods.
- Throughout the paper, the authors do not specify the parameter size of InternVL2 nor provide any relevant citation, which limits the reproducibility and clarity of their comparisons. Additionally, InternVL2 is currently not the most state-of-the-art open-source model; therefore, including a comparison with the latest state-of-the-art visual models would significantly strengthen the experimental validity.
- Although PRIMITIVE demonstrates promising performance and improved interpretability in Figures 14 and 15, the authors have not adequately addressed the reasoning errors that still persist in problem-solving scenarios. These errors may not necessarily stem from perceptual inaccuracies but could potentially result from hallucinations within the MLLMs' reasoning processes. It would be beneficial to include additional experiments or analyses in caption-based tasks to better understand and address these reasoning limitations.

**Methods And Evaluation Criteria:**

- The introduction of GeoGLIP as a specialized visual encoder effectively addresses the challenge of accurately detecting small-scale geometric primitives, significantly advancing the state-of-the-art in math-oriented visual understanding.
- The proposed feature router mechanism is innovative, enabling adaptive fusion of multi-scale visual information from GeoGLIP and CLIP, thus demonstrating significant empirical improvements.
- The technical contributions of the proposed PRIMITIVE pipeline appear limited, as the approach primarily integrates existing LLM architectures with incremental modifications.

**Other Comments Or Suggestions:**

N/A

**Other Strengths And Weaknesses:**

- While the authors extensively evaluate their method on mathematical visual reasoning tasks, additional qualitative analyses or visualization examples illustrating the feature routing decisions and specific cases where PRIMITIVE significantly outperforms baselines could further clarify the interpretability and practical effectiveness of the dynamic feature routing mechanism.
- Figure 3 and Table 5 extend beyond the page margins, which compromises the visual presentation and overall readability.

**Questions For Authors:**

N/A

**Relation To Broader Scientific Literature:**

N/A

**Theoretical Claims:**

N/A

---

> ### Author Rebuttal · Authors · 2025-04-01
>
> # We thank the reviewer for insightful questions that help refine our work further.
>
> ## 1. The impact of visual encoders—e.g., comparing Qwen2.5-VL and PRIMITIVE-Qwen2.5-7B with fixed LLM weights.
>
> To address the concern about visual encoders' impact on reasoning, we compare variants in controlled settings, as detailed in lines 840–847 and Tab. 6 of the appendix.
>
> Although Qwen2.5-VL and PRIMITIVE-Qwen2.5-7B share the same LLM backbone, they differ significantly in training recipes and dataset scale (~2T vs. ~600K). For details, see Q2 (Ay8j). In Tab. 6, we control for training settings and datasets, isolating the visual encoder for direct comparison.  See Q1 (sjVj) for analysis.  For qualitative response differences, we analyzed model reasoning steps and found that the GLIP+CLIP dual encoder struggles with basic shape perception and interrelationship descriptions, often leading to hallucinations and incorrect answers. We will include their rationale demos in the revised paper.
>
> To ensure a fair comparison with Qwen2.5-VL, we conducted new experiments resulting in Qwen2.5-VL-7B+ and PRIMITIVE-Qwen2.5-VL-7B (see Q2 for ARAy8j), with more detailed comparisons to be added in the revision.
>
> ## 2. The PRIMITIVE pipeline seems limited in contribution, as it mainly integrates existing LLMs with minor modifications.
>
> We respectfully disagree that our contributions are merely incremental. Our core contribution is the design of a geometrically sensitive visual encoder that enhances MLLMs' fine-grained primitive perception—addressing a key bottleneck in reasoning, as shown by the analysis in Figs. 1 and 5a.
>
> Unlike single-modality LLMs, MLLMs rely on accurate visual cue interpretation to support abstract reasoning. Misinterpretations at this stage can misguide reasoning processes. To address this, we designed a geometrically sensitive visual encoder with box- and pixel-level supervision, going beyond the image-level supervision used in CLIP (see Q3 for AR PhN5). This was enabled by a custom data engine generating diagrams with shape, junction, and boundary annotations.
>
> Furthermore, we integrated the trained visual encoder into LLMs using two methods: hard coordinates (Sec. A.4) and soft visual prompts. For the latter, we introduced a feature router that dynamically selects features from semantic- to geometric-rich levels. Our designs were implemented in LLaMA2-7B, Deepseek-math-7B, and Qwen-math-7B, with ablations showing notable gains over baselines across three math reasoning benchmarks.
>
> ## 3. The paper lacks the parameter size and citation for InternVL2, limiting clarity and reproducibility; adding comparisons with recent state-of-the-art visual models would strengthen the experimental validity.
>
> We apologize for the oversight—InternVL2 has 8B parameters, which we will clarify as InternVL2-8B. For reproducibility, we will release model weights for PRIMITIVE-7B, PRIMITIVE-Deepseek-7B, and PRIMITIVE-Qwen2.5-7B, along with training and inference code. To address comparisons with state-of-the-art models, we have integrated our design into Qwen2.5-VL (see Q2 for ARAy8j).
>
> ## 4.  It lacks analysis of remaining reasoning errors, which may stem from LLM hallucinations rather than perception.
>
>  Our primary goal is to enhance the visual grounding of MLLMs to complement their reasoning, not to solve the full spectrum of mathematical reasoning tasks. We acknowledge that hallucinations in reasoning remain an important challenge, influenced by many factors such as data distribution, vision encoders, modality alignment, and the LLM's inherent knowledge [1–5].
>
> Perceptual enhancement is one feasible path to reducing such hallucinations [4,5], and our work focuses on improving object- and pixel-level perception in mathematical contexts. We agree that other factors also impact reasoning accuracy and plan to explore them in future work. As suggested, analyzing caption-based descriptions could provide intuitive insights into hallucinations tied to diagram elements. However, building such datasets and benchmarks is non-trivial and beyond the scope of this rebuttal, but it remains a key direction for future research.
>
> [1] Jaemin Cho et al., Fine-grained image captioning with clip reward.
>
> [2]  Ziwei Ji et al., Survey of hallucination in natural language generation.
>
> [3]  Lei Huang et al., A survey on hallucination in large language models: Principles, taxonomy, challenges, and open questions.
>
> [4]  Shunqi Mao et al., Through the Magnifying Glass: Adaptive Perception Magnification for Hallucination-Free VLM Decoding.
>
> [5]  Hanchao Liu et al., A Survey on Hallucination in Large Vision-Language Models.
>
> ## 5.  Additional visualization examples are needed, and Fig. 3 and Tab. 5 exceed page margins.
>
> Thank you. We will add more qualitative comparisons between PRIMITIVE and baselines for clarity and fix the formatting issues in Fig. 3 and Tab. 5.

---

> > ### Comment · Reviewer_nRX6 · 2025-04-05
> >
> > Thanks for your response. The response partially addressed my concerns. I would like to raise my rating.

---

> > > ### Author Response · Authors · 2025-04-06
> > >
> > > Esteemed Reviewer,
> > >
> > > Thank you for your kind message, and valuable comments helping us improve and refine our manuscript. We are glad to hear that our responses have addressed some of your concerns. If you have any further questions or suggestions or would like to discuss some further issues, kindly do let us know how we can resolve them. We are very keen to engage with yourself and improve our work further.
> > >
> > > Kind regards,
> > >
> > > The Authors

---

### Official Review · Reviewer_Ay8j · 2025-03-14

**Overall Recommendation:** 3

**Summary:**

This work proposes a novel approach named PRIMITIVE, aiming to address the deficiencies of current mathematical multimodal large language models (MLLMs) in geometric perception, thereby enhancing their capabilities in visual mathematical reasoning. Experiments conducted on three benchmarks demonstrate the effectiveness of the PRIMITIVE method.

## Update review

In the previous round, I thought the feature pyramid used by the authors lacked novelty. The authors replied that they did not consider the feature pyramid technique as a major contribution. I hope the authors can clarify this in the revision.

Additionally, I still have concerns about the authors using existing models to extract junctions, which may introduce unpredictable noise, as the performance on MathVerse and MathVista is not very promising. I hope the authors will address this point in the revised version.

The study of how fine-grained visual perception influences downstream reasoning in MLLMs is meaningful. Therefore, I change my opinion to 'Weak Accept'.

**Claims And Evidence:**

No, please refer to Part of [Methods And Evaluation  Criteria]

**Essential References Not Discussed:**

N/A

**Experimental Designs Or Analyses:**

- 1) The PRIMITIVE architecture used in this work is built upon LLaVA-1.5, which is somewhat outdated. Moreover, PRIMITIVE employs a feature pyramid structure to extract fine-grained geometric visual features, a method that is generic for natural images and not specifically tailored for geometric image tasks.

**Methods And Evaluation Criteria:**

- 1) The authors mentioned that existing models would be used to extract junctions and boundaries as ground truth. For geometric shapes with diverse variations in shapes and domains, could this process introduce noise?

- 2) The performance on MathVerse and MathVista is not very promising. For example, the proposed PRIMITIVE-Qwen2.5-7B achieves lower performance on the MathVista evaluation set compared to InternVL2.5-8B (64.4%), Qwen2.5-VL-7B (68.2%), and even previous-generation MLLMs such as InternVL2-8B (61.6%), InternVL2-4B (57.0%), and Qwen2-VL-7B (58.2%).

- 3) The relatively low experimental results on the MathVista and MathVerse evaluation sets raise doubts about the effectiveness of the proposed approach. Although the work claims that the approach can be integrated into multiple model baselines, the experimental results for its combination with baselines such as Qwen2.5-7B and DeepSeek-7B remain unsatisfactory.

**Other Comments Or Suggestions:**

The figure captions should be placed below the figures. Please make this correction in future versions.

**Other Strengths And Weaknesses:**

[Strengths]

- 1) This work analyzes the proportion of perception errors in geometric question answering, which is insightful.

- 2) The authors achieved high performance on the GeoQA leaderboard.

[Weaknesses]

- 1) The authors mentioned that existing models would be used to extract junctions and boundaries as ground truth. For geometric shapes with diverse variations in shapes and domains, could this process introduce noise?

- 2) The performance on MathVerse and MathVista is not very promising. For example, the proposed PRIMITIVE-Qwen2.5-7B achieves lower performance on the MathVista evaluation set compared to InternVL2.5-8B (64.4%), Qwen2.5-VL-7B (68.2%), and even previous-generation MLLMs such as InternVL2-8B (61.6%), InternVL2-4B (57.0%), and Qwen2-VL-7B (58.2%).

- 3) The relatively low experimental results on the MathVista and MathVerse evaluation sets raise doubts about the effectiveness of the proposed approach. Although the work claims that the approach can be integrated into multiple model baselines, the experimental results for its combination with baselines such as Qwen2.5-7B and DeepSeek-7B remain unsatisfactory.

- 4) The PRIMITIVE architecture used in this work is built upon LLaVA-1.5, which is somewhat outdated. Moreover, PRIMITIVE employs a feature pyramid structure to extract fine-grained geometric visual features, a method that is generic for natural images and not specifically tailored for geometric image tasks.

**Questions For Authors:**

N/A

**Relation To Broader Scientific Literature:**

This work proposes MLLMs in mathematical scenarios, which could potentially aid in analyzing the reasoning capabilities of MLLMs.

**Theoretical Claims:**

This research work does not propose theoretical analyses.

---

> ### Author Rebuttal · Authors · 2025-04-01
>
> # We thank the reviewer for insightful questions that help refine our work further.
>
> ## 1.  The authors use existing models to extract junctions and boundaries as ground truth—could this introduce noise given the variation in geometric shapes and domains?
>
> For junction detection, we discuss noisy cases in lines 802–807 of Appendix A.4 and show failure cases in Fig. 11. We use a CNN model (Huang et al., 2018) trained on Man-Made Environments to generate ground truth. Some label noise arises from out-of-domain (OOD) diagrams, which we mitigated using a 0.85 confidence threshold and manual correction. While 100\% accuracy isn’t guaranteed across 20K+ samples, the noise level is acceptable. The test set recall is 85.6%, and GeoGLIP remains robust to distortions (see Q2 for AR sjVj). Improving junction labeling and data synthesis, as discussed in Sec. A.7, could further boost performance.
>
> For boundary detection, we use the FoJ model (Verbin & Zickler, 2021), which applies a machine learning algorithm for generalized M-junctions. Unlike CNNs, FoJ generalizes better to OOD domains and is less dataset-biased. Its noise resilience is supported by prior work. GeoGLIP achieves an IoU of 92.3% and remains robust under various distortions (see Q2 for AR sjVj).
>
> ## 2. PRIMITIVE-Qwen2.5-7B shows lower  MathVerse/MathVista performance compared to Qwen2.5-VL-7B (68.2%), and even earlier MLLMs.
>
> MLLMs fall into two categories: generic (e.g., InternVL2.5, Qwen2.5-VL, GPT4o) and math-specific (e.g., G-LLaVA, Math-LLaVA, MAVIS, ours), differing significantly in training recipes and datasets when integrated with LLMs and visual encoders.
>
> Generic models are trained on large-scale visual instruction datasets covering a wide variety of multimodal data, including OCR, academic questions, localization data, documents, and video descriptions ( e.g., Qwen2.5-VL uses 2TB+ data), while math-specific models utilize only mathematical text-diagram pairs, which require fewer training resources and provide an efficient test base for proposed math module designs. Our model is fairly compared with math-specific MLLMs, as it is trained solely on MathV360K+Geo170K. In controlled experiments, adding GeoGLIP and the soft router boosts baseline G-LLaVA by 4.6% on MathVerse and 12.3% on MathVista. We've also extended our approach to DeepSeek-Math-7B and Qwen2.5-Math-7B, achieving consistent +6% gains on MathVista (Tab. 7). Note: baselines use only CLIP visual encoders and the same training setup—not generic DeepSeek-VL or Qwen2.5-VL.
>
> To compare with state-of-the-art 7B models, we implemented our designs on Qwen2.5-VL-7B and fine-tuned it using its official checkpoint. As the training code is not publicly available, significant effort was required to rewrite the functions, data loader, and training processes within the transformers and Hugging Face frameworks. For alignment, we trained only the projectors;  Due to time limits, we trained both projectors and LLMs using LoRA to accelerate SFT stage on the MathV360K+Geo170K dataset.
>
> Under the same training setup, we developed two versions: (1) Qwen2.5-VL-7B+, fine-tuned with math-specific visual data; and (2) PRIMITIVE-Qwen2.5-VL-7B, integrated with our GeoGLIP and soft router. Our integrated models are lightweight (<50MB, see Sec. A.6.2) and already show performance gains. We plan to further improve results by exploring full fine-tuning (beyond LoRA) and partial unfreezing of the visual encoder with lower learning rates.
>
>
> |Model|MathVista/All (acc)|MathVerse/All (acc)|
> |:-:|:-:|:-:|
> |Qwen2.5-VL-7B|68.2|49.2|
> |Qwen2.5-VL-7B+|68.5|49.8|
> |PRIMITIVE-Qwen2.5-VL-7B|69.7|51.0|
>
> ## 3. PRIMITIVE uses a feature pyramid to extract fine-grained features—a technique common in natural images but not specifically tailored for geometric tasks.
>
> Thank you for your observations regarding the PRIMITIVE architecture. Although the feature pyramid structure effectively extracts fine-grained visual cues in natural images, it is not central to enhancing mathematical perception in our MLLM.  Instead, the key factor lies in the mathematically sensitive GeoGLIP, which incorporates visual-centric mathematical training datasets and fine-grained box- and pixel-level supervision (refer to Q3 for AR PhN5).
>
> Our empirical evidence, detailed in lines 840-848 and Tab. 6 of the appendix, supports this. Specifically, coupling the original GLIP (trained on natural images) with the feature pyramid and soft router techniques decrease performance from 67.0 to 65.3 on GeoQA. Additionally, Fig. 10 (right panel) demonstrates GLIP's inability to accurately perceive basic geometric primitives.
>
> These results confirm that fine-grained primitive perception ability depends on a geometrically sensitive visual encoder, and then this ability could be further enhanced by employing pyramid techniques. Our specific adaptations and innovations for GeoGLIP are deliberately designed to tackle the unique challenges of mathematical image tasks.

---

> > ### Comment · Reviewer_Ay8j · 2025-04-02
> >
> > Thank you for the authors' feedback. After reading the rebuttal, I still have concerns regarding the novelty of this work, such as the approach of combining LlaVA-1.5 with the feature pyramid, as well as the GeoGLIP performing shape grounding, boundary, and junction detection. I believe there may be potential noise in the predictions for these aspects. Therefore, I maintain my initial score.

---

> > > ### Author Response · Authors · 2025-04-02
> > >
> > > ## Thank you for your thoughtful feedback and for engaging with our rebuttal.
> > >
> > > We address reviewer's two main concerns regarding novelty and potential prediction noise in GeoGLIP.
> > >
> > > # 1.  Novelty and Contribution:
> > > Our key contribution lies in identifying fine-grained visual perception as a major bottleneck in current MLLMs' reasoning ability (see Figs. 1 and 5a). To address this, we design a geometrically sensitive visual encoder and enhance perception through fine-grained supervision—using bounding box, junction, and boundary labels generated by a custom data engine. We instantiate our design (GeoGLIP + global feature router) across LLaMA2-7B, Deepseek-Math-7B and Qwen-Math-7B. This improves reasoning by enhancing visual grounding, achieving +6.4\% to +12.3\% over baselines on MathVista. Our method focuses on perception enhancement, which is orthogonal to existing methods focused on improving reasoning, and we believe this opens new directions for mathematical MLLM research.
> > >
> > > We do not claim the feature pyramid as a novel contribution; it is used to enhance perception within our global feature router, enabling adaptive selection from geometric- to semantic-rich cues to aid reasoning. As noted in Q3, without a geometry-sensitive visual encoder, the feature pyramid alone cannot effectively extract fine-grained mathematical primitives.
> > >
> > > # 2. Prediction Noise:
> > > We address the reviewer’s concern along three parts:
> > >
> > > * **Robustness of GeoGLIP to Visual Distortions**:
> > > As shown in Q2 (AR sjVj), GeoGLIP is robust to common visual distortions such as Gaussian noise, changes in line style, line width, and resolution.  Compared to the baseline CLIP visual encoder, GeoGLIP shows much smaller performance degradation under these distortions. This robustness is attributed to our diverse training data and augmentation strategies, including randomly varying line width (1–5), flipping, cropping, and keep-ratio resizing. Moreover, it's important to note that even models trained with perfect ground truth labels cannot guarantee 100\% inference accuracy. For example, in natural image object detection, no model achieves 100\% mAP on standard benchmarks like COCO.
> > >
> > > * **Design Choices to Minimize the Impact of Prediction Uncertainty**:
> > > As stated in the introduction (lines 74–77): "Given GeoGLIP’s inherent uncertainty in detecting geometric primitives, instead of directly prompting LLMs with primitive locations (e.g., hard coordinates; see Sec. A.4 for ablation), we leverage global pyramid feature maps that encode essential information for pixel-to-shape detection." In other words, we do not directly use prediction results as hard prompts for the LLM; instead, we use global feature maps as soft inputs. This design choice keeps our model lightweight—under 50MB—as it requires only a visual encoder without additional detection heads (see Sec. A.6.2).
> > >
> > > * **Focus of This Work**:
> > > This work does not aim to develop a standalone primitive detector. Instead, we focus on how fine-grained visual understanding influences downstream reasoning in MLLMs. As shown in Fig. 2, our model generates more fine-grained attention maps, effectively highlighting not only boundaries but also detailed elements like dotted lines and right-angle symbols.
> > >
> > > We would greatly appreciate any suggestions from the reviewer on how to design future experiments to better address your concerns.

---

### Official Review · Reviewer_PhN5 · 2025-03-20

**Overall Recommendation:** 3

**Summary:**

This paper proposes PRIMITIVE, a multi-modal large language model for mathematical problem-solving. The contribution of PRIMITIVE is two-folds: a mathematical vision encoder, GeoGLIP, and an MLP-based feature router. The GeoGLIP is pre-trained using synthetic data with box-level and pixel-level loss within the GLIP architecture. The feature router adopts a soft weight strategy to combine different levels of visual feature adaptively. The performance of PRIMITIVE is good on different benchmarks.

**Claims And Evidence:**

Yes

**Essential References Not Discussed:**

The main motivation and training approaches of this paper are similar to MAVIS, e.g., improving vision encoder and three-stage training, although with different techniques. I suggest discuss more about the relation and difference of this paper and MAVIS in the Intro and Related Work parts.

**Experimental Designs Or Analyses:**

It is not very fair for performance comparison. In section 4.1, the authors mention that they evaluate PRIMITIVE on GeoQA and MathVerse/MathVista separately using different checkpoints, that is, Geo170K and MathV360K, respectively. This is my main of concern, since previous works in this field or the default choice of this community, is to test all different benchmarks using one trained model, either by combining training data in one stage or setting different training stages. If PRIMITIVE cannot perform well on different benchmarks within one checkpoint, the generalization capability of this approach will be questioned. The author is encouraged to provide experiments or illustrations.

**Methods And Evaluation Criteria:**

Yes

**Other Comments Or Suggestions:**

Overall it's a good paper, while remains some issues to be solved.

**Other Strengths And Weaknesses:**

Strengths:

1. Exploring the role of vision encoder in math problem-solving is interesting and reasonable, which is mostly ignored by existing research. Although a few works in recent months also focus on this point, this paper provides unique insights with localization loss.

2. The curated datasets for pre-training GeoGLIP is useful to the community.

3. The figure is clearly presented.

Weaknesses:

1. Please refer to the experimental fairness above.

2. The title of the paper is a bit overclaimed. PRIMITIVE utilizes localization information to enhance the vision perception and performance, but does not enable LLMs to know where to focus within math figures.

**Questions For Authors:**

no

**Relation To Broader Scientific Literature:**

No concern

**Theoretical Claims:**

The formula (1) in Section 3.3 should contain more comprehensive illustration of newly defined labels.

---

> ### Author Rebuttal · Authors · 2025-04-01
>
> # We thank the reviewer for insightful questions that help refine our work further.
> ## 1. The formula (1) in Section 3.3 should contain more comprehensive illustration of newly defined labels.
>
> Thank you. Duty noted. Eq. (1) describes the soft router process. The scalar routing weight for each level is denoted as $w^i$, where $i \in \{1, 2, 3, 4\}$; the symbol $\circledcirc$ denotes the operation flow from right to left. For instance, $\mathcal{G} \circledcirc F_{\text{geo}}^i $ indicates that the different levels of GeoGLIP features $\(F_{\text{geo}}^i\)$ are resized $(\mathcal{G})$, which is essential for aligning spatial size of CLIP features.  $\sigma$ denotes the normalization function, which could be either a SoftMax or a Sigmoid function, depending on the feature integration process (channel-wise or sequence-wise concatenation).
>
> We have  added a more detailed description of each component and its role in the revised manuscript to ensure that the formulation and its application are transparent to the readers.
>
> ## 2. In Section 4.1, authors use two checkpoints for evaluation on GeoQA and MathVerse/MathVista, which is not considered fair for performance comparison since the standard practice in this field is to test all benchmarks using a single trained model.
>
>
> Thank you for highlighting this important point. GeoQA primarily evaluates geometric problem reasoning, and our baseline G-LLaVA-7B is trained exclusively on the Geo170K dataset for this benchmark. To ensure a fair comparison, we maintain consistency in the training dataset and test on GeoQA.
>
> Regarding the MathVista and MathVerse benchmarks, which cover a diversity of subjects including scientific topics, PaperQA, and IconQA, we applied the additional MathV360K dataset for fine-tuning our model (Geo170K+MathV360K), a combination commonly used by other mathematical MLLMs like Math-LLaVA and MAVIS. In response to this concern, we test our model trained on Geo170K+MathV360K on the GeoQA benchmark. The performance improved, with PRIMITIVE-7B's top-1 accuracy increasing to 71.3 (67.0 in Tab. 3). We adopt this version as the default choice and will release the checkpoint to the public.
>
> ## 3.  The main motivation and training approaches of this paper are similar to MAVIS, e.g., improving vision encoder and three-stage training, although with different techniques. I suggest discuss more about the relation and difference of this paper and MAVIS in the Intro and Related Work parts.
>
> Thank you for your suggestion. We have expanded the discussion regarding our model in comparison to MAVIS in the Related Works section.
>
> Although our high-level motivation and MLLM training strategies share similarities with MAVIS, our different training techniques for the visual encoder and MLLM design enable our model to achieve comparable reasoning performance using an 8$\times$ smaller visual instruction training dataset.  For visual encoder training, MAVIS utilizes 588K caption-diagram pairs with alignment loss similar to CLIP, which provides image-level supervision. In contrast, our model employs only 40K samples with box-level and pixel-level supervision, offering more detailed local feature perception.  Researchers have shown that CLIP-style encoders  fail to capture fine details (e.g., Yiwu Zhong et al., "RegionCLIP: Region-based Language-Image Pretraining"; Zhe Gan et al., "Vision-language pre-training: Basics, recent advances, and future trends").
>
> A more intuitive demonstration is shown through the visualization of attention maps. MAVIS's visual encoder, depicted in Fig. 1(a) of MAVIS's paper, primarily highlights coarse-level boundary information and often overlooks detailed features. In contrast, our model, as shown in Fig. 2, produces attention maps that are more fine-grained, clearly highlighting not only the boundaries but also detailed elements such as dotted lines and right-angle symbols. This stark contrast underscores the superior local feature detection capability of our model.  Furthermore, our model design incorporates a soft router that adaptively selects visual cues ranging from semantic-rich to geometric-rich. This adaptability enhances the model's ability to accurately solve problems by leveraging relevant visual information effectively.
>
> ## 4. The title of the paper is a bit overclaimed. PRIMITIVE utilizes localization information to enhance the vision perception and performance, but does not enable LLMs to know where to focus within math figures.
>
> Thank you. Duty noted.  We will revise the title to more accurately reflect the model's enhanced visual perception and avoid overclaiming.

---

### Decision · Program_Chairs · 2025-05-01

**Decision:**

Accept (poster)

**Comment:**

The paper identified problems in fine-grained visual understanding for math VQA, and proposed a new encoder and a routing method to address it. The paper received unanimous positive ratings (3, 3, 3, 4). The problem is well-motivated, and the empirical evidence are strong for the proposed improvement, according to the reviewers. The AC recommends acceptance.